EMBO
Molecular Medicine

# Dual IRE1 RNase functions dictate glioblastoma development

Stéphanie Lhomond[1,†], Tony Avril[2,3,†], Nicolas Dejeans[1,‡], Konstantinos Voutetakis[4,5,‡], Dimitrios Doultsinos[2,3], Mari McMahon[2,3,6], Raphaël Pineau[2,3], Joanna Obacz[2,3], Olga Papadodima[4], Florence Jouan[2,3], Heloise Bourien[2,3], Marianthi Logotheti[4,7], Gwénaële Jégou[2,3], Néstor Pallares-Lupon[1], Kathleen Schmit[1], Pierre-Jean Le Reste[2,8], Amandine Etcheverry[9], Jean Mosser[9], Kim Barroso[2,3], Elodie Vauléon[2,3], Marion Maurel[2,3,6], Afshin Samali[6] (iD), John B Patterson[10], Olivier Pluquet[11], Claudio Hetz[12,13,14,15,16], Véronique Quillien[2,3], Aristotelis Chatziioannou[4,7] & Eric Chevet[2,3,*] (iD)

## Abstract

Proteostasis imbalance is emerging as a major hallmark of cancer, driving tumor aggressiveness. Evidence suggests that the endoplasmic reticulum (ER), a major site for protein folding and quality control, plays a critical role in cancer development. This concept is valid in glioblastoma multiform (GBM), the most lethal primary brain cancer with no effective treatment. We previously demonstrated that the ER stress sensor IRE1α (referred to as IRE1) contributes to GBM progression, through XBP1 mRNA splicing and regulated IRE1-dependent decay (RIDD) of RNA. Here, we first demonstrated IRE1 signaling significance to human GBM and defined specific IRE1-dependent gene expression signatures that were confronted to human GBM transcriptomes. This approach allowed us to demonstrate the antagonistic roles of XBP1 mRNA splicing and RIDD on tumor outcomes, mainly through selective remodeling of the tumor stroma. This study provides the first demonstration of a dual role of IRE1 downstream signaling in cancer and opens a new therapeutic window to abrogate tumor progression.

**Keywords** cancer; endoplasmic reticulum; IRE1; regulated IRE1-dependent decay; XBP1

**Subject Categories** Cancer; Chromatin, Epigenetics, Genomics & Functional Genomics; Neuroscience

## Introduction

Glioblastoma multiforme (GBM) is one of the most lethal adult cancers, as the majority of patients die within 15 months after diagnosis (Anton *et al*, 2007). GBM is an aggressive, incurable glioma (grade IV astrocytoma, WHO classification) due to great heterogeneity of cell subtypes within the tumor and to the presence of invasive spots that cannot be easily cured by surgical resection or targeted radiation. To limit tumor recurrences from invasive cells, chemotherapy [temozolomide (TMZ)] was added to surgery and radiation (Stupp *et al*, 2005). Although this combined therapy has demonstrated some efficiency, it only increases patient's median

---

1  Université de Bordeaux, Bordeaux, France
2  INSERM U1242, "Chemistry, Oncogenesis, Stress, Signaling", Université de Rennes 1, Rennes, France
3  Centre de Lutte Contre le Cancer Eugène Marquis, Rennes, France
4  Institute of Biology, Medicinal Chemistry & Biotechnology, NHRF, Athens, Greece
5  Department of Biochemistry & Biotechnology, University of Thessaly, Larissa, Greece
6  Apoptosis Research Centre, School of Natural Sciences, NUI Galway, Galway, Ireland
7  e-NIOS PC, Kallithea-Athens, Greece
8  Department of Neurosurgery, University Hospital Pontchaillou, Rennes, France
9  Integrated Functional Genomics and Biomarkers Team, UMR6290, CNRS, Université de Rennes 1, Rennes, France
10 Medinnovata Inc., Ventura, CA, USA
11 Institut Pasteur de Lille, CNRS UMR8161 "Mechanisms of Tumourigenesis and Targeted Therapies", Université de Lille, Lille, France
12 Biomedical Neuroscience Institute, Faculty of Medicine, University of Chile, Santiago, Chile
13 Program of Cellular and Molecular Biology, Institute of Biomedical Sciences, University of Chile, Santiago, Chile
14 Center for Geroscience, Brain Health and Metabolism, Santiago, Chile
15 Buck Institute for Research on Aging, Novato, CA, USA
16 Department of Immunology and Infectious diseases, Harvard School of Public Health, Boston, MA, USA
   *Corresponding author. Tel: +33 223237258; E-mail: eric.chevet@inserm.fr
   †These authors contributed equally to this work as first authors
   ‡These authors contributed equally to this work as second authors

survival from 12.1 to 14.6 months. Thus, understanding biological processes of GBM progression and treatment resistance represents a major challenge to develop more effective therapies.

The ER is the major subcellular compartment involved in protein folding and secretion. Accumulating evidence supports an emerging role of ER proteostasis alterations in cancer development, having been implicated in most hallmarks of cancer (Urra *et al*, 2016). ER stress triggers an adaptive reaction known as the unfolded protein response (UPR), which aims to recover proteostasis or to induce apoptosis of irreversibly damaged cells (Walter & Ron, 2011). Several studies in animal models of cancer using genetic or pharmacological manipulation of the UPR have demonstrated a functional role of this pathway in cancer (Hetz *et al*, 2013). The UPR is initiated by the activation of three ER transmembrane proteins known as PERK, ATF6, and IRE1 (Hetz *et al*, 2015). IRE1α (referred to as IRE1 hereafter) is a serine/threonine kinase and endoribonuclease that represents the most conserved UPR signaling branch in evolution, controlling cell fate under ER stress (Hetz *et al*, 2015). Once activated, IRE1 oligomerizes thus engaging three major downstream outputs including the activation of JNK (Urano *et al*, 2000; Han *et al*, 2009), the splicing of XBP1 mRNA (XBP1s) (Yoshida *et al*, 2001; Calfon *et al*, 2002), and the degradation of targeted mRNA and miRNA, a process referred to as RNA regulated IRE1-dependent decay (RIDD) (Maurel *et al*, 2014). Importantly, the universe of RIDD targets may depend on the tissue context and the nature of the stress stimuli, impacting different biological processes including apoptosis, cell migration, and inflammatory responses (Dejeans *et al*, 2014). Several functional studies have shown that targeting the expression or the RNase activity of IRE1 reduces the progression of various forms of cancer mostly due to ablating the prosurvival effects of XBP1 on tumor growth (Chevet *et al*, 2015; Obacz *et al*, 2017), and we have previously demonstrated its functional implication in various models of experimental glioblastoma (Drogat *et al*, 2007; Auf *et al*, 2010; Dejeans *et al*, 2012; Pluquet *et al*, 2013; Jabouille *et al*, 2015). Moreover, large-scale sequencing studies on human cancer tissue samples performed by The Cancer Genome Atlas (TCGA) initiative (Cancer Genome Atlas Research Network, 2008; Parsons *et al*, 2008) revealed the presence of three somatic mutations on the IRE1 gene in GBM leading to the S769F, Q780* (Greenman *et al*, 2007), and P336L (Parsons *et al*, 2008) variants. Although a previous report aimed at understanding the structural impact of some of those mutations in IRE1 function (Xue *et al*, 2011), little is known on how their differential contribution to RIDD and XBP1 mRNA splicing impacts on GBM development and progression.

Our previous findings indicated that IRE1 also contributes to mRNA degradation in cancer, having unexpected roles in tissue invasion in GBM, in addition to affecting growth and vascularization (Dejeans *et al*, 2012; Pluquet *et al*, 2013). Here, we took advantage of the selective signaling properties of different IRE1 GBM somatic mutants and we demonstrate that the modulation of IRE1 signaling characteristics in GBM cells controls tumor aggressiveness, not only by providing selective advantages to the tumor cells themselves, but also by remodeling the tumor stroma to the benefit of growth. Furthermore, we provide evidence supporting a novel concept where IRE1-downstream signals play antagonistic roles in cancer development, where XBP1s provides pro-tumoral signals, whereas RIDD of mRNA and miR17 rather elicits anti-tumoral features. Our

data, obtained using established cell lines, patient tumor samples, and primary GBM lines, depict a complex scenario where IRE1 signaling orchestrates distinct aspects of GBM biology, thereby offering novel targets for therapeutic intervention.

## Results

### IRE1 activity and human GBM tumor properties

We previously identified an IRE1-dependent gene expression signature in U87 cells using IRE1 dominant-negative-expressing cells, an approach that fully blocks all RNase outputs of this ER stress sensor (Pluquet *et al*, 2013). Functional annotation of the genes comprised in the IRE1-dependent gene expression signature revealed the enrichment in biological functions associated with stress responses, cell adhesion/migration, and with the inflammatory and immune response (Fig 1A). This gene expression signature was processed through the Bioinfominer pipeline (Appendix Fig S1) to increase its functional relevance, and this led to the identification of 38 IRE1 signaling hub genes (Appendix Fig S1). This 38 genes signature was then confronted to the transcriptomes of the GBM TCGA (Cancer Genome Atlas Research Network, 2008) and GBMmark (in-house) cohorts (Fig 1B). This analysis revealed the existence of two populations of patients displaying either high or low IRE1 activity, respectively (Fig 1C and Appendix Fig S2). Tumors exhibiting high IRE1 activity also correlated with shorter survival of the corresponding patients (Fig 1D). We then tested the impact of IRE1 signaling on the expression levels of IBA1, CD14, and CD163 as markers of the inflammatory/immune response in the tumors (Fig 1E), the levels of CD31 and vWF to monitor angiogenesis (Fig 1F), or RHOA, CYR61, and CTGF expression as indicators of tumoral invasion (Fig 1G). This revealed that tumors exhibiting high IRE1 activity also presented markers of massive infiltration of macrophages, with high vascularization and invasive properties. Similar observations were also obtained when analyzing the GBMmark dataset (Appendix Fig S2B–D). Activation of the IRE1/XBP1 axis was confirmed in those tumors through the analysis of the expression of XBP1 target genes ERDJ4 and EDEM1 (Appendix Fig S2E). To confirm these observations at the protein level in GBM, fresh tumors presenting high or low IRE1 activity were dissociated and analyzed for CD45 and CD11b expression by FACS. This analysis revealed that high IRE1 signaling correlated with strong macrophage infiltration (Fig 1H). Moreover, the presence of endothelial cells in tumors was detected by FACS after CD31 labeling and was increased in GBM tumors exhibiting high IRE1 activity (Fig 1I). Finally, tumors exhibiting high IRE1 signaling were mainly classified as belonging to the mesenchymal type of GBM whereas those with low IRE1 activity mostly included pro-neural and classical tumors (Fig 1J). These data demonstrate that IRE1 activation is found in human tumors and correlate with more aggressive cancers with shorter patient survival.

### Identification of a novel somatic mutation on IRE1 in human GBM

IRE1 activation in tumors could be due to exposure to stressful environments (nutrient/oxygen deprivation, pH, immune response) but

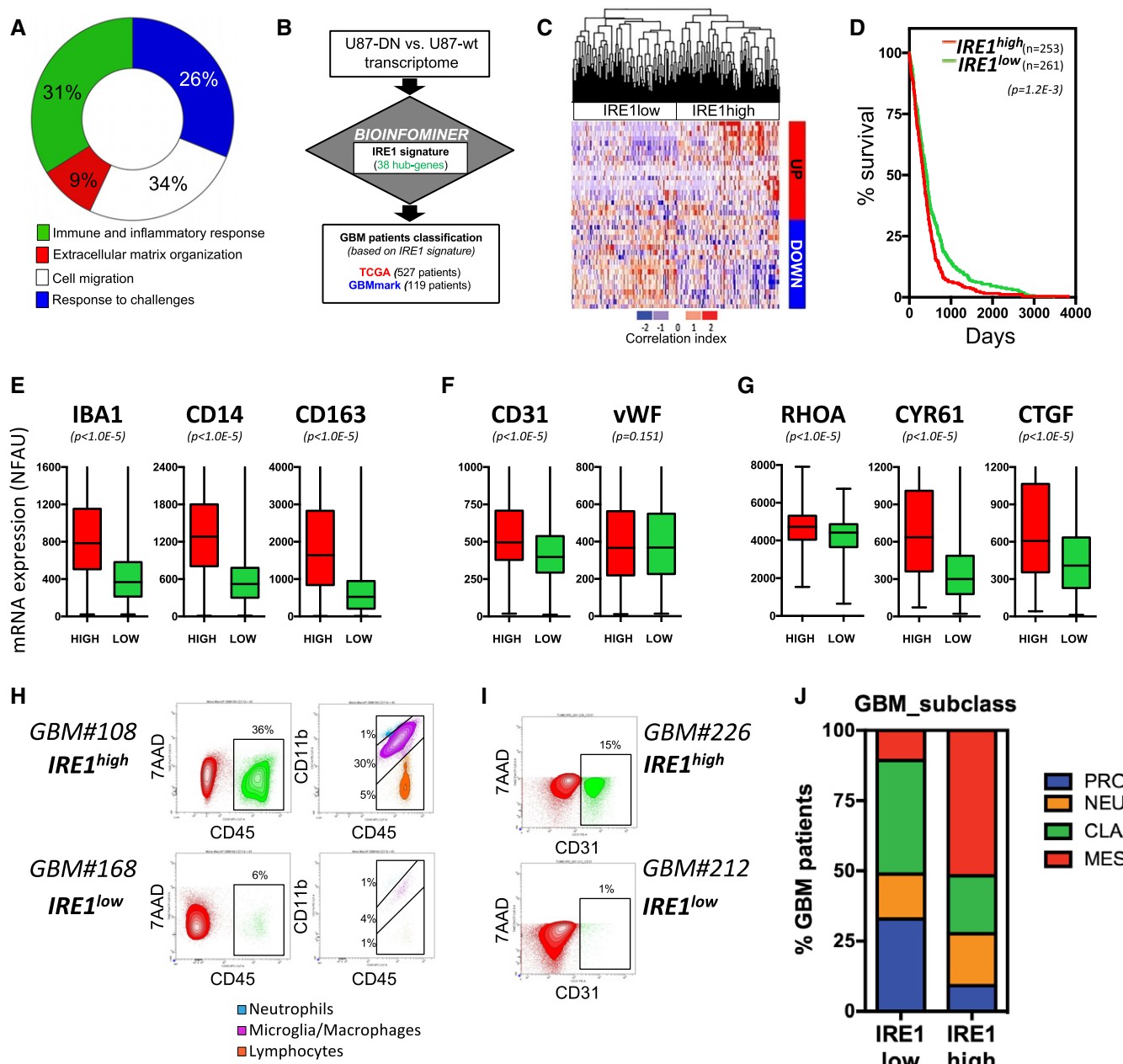

**Figure 1. IRE1 signaling signatures in glioblastoma multiform.**

A    Functional annotations of the IRE1 gene expression signature identified in U87 cells (Pluquet *et al*, 2013).

B    Schematic representation of the analysis workflow.

C    Hierarchical clustering of GBM patients (TCGA cohort) based on high or low IRE1 activity as assessed with the correlation index of their median *z*-score with the expression pattern of the IRE1 gene signature of 38 hub genes (see Materials and Methods and Appendix Fig S1). Pearson correlation was used to measure the similarity between different genes and tumor cases, as well. The correlation index refers to the gene expression median *z*-score with (+) or (−) sign for identical or reverse expression pattern with that of WT vs. DN, respectively. The expression pattern of WT vs. DN has been described in detail in Pluquet *et al* (2013). Blue: low correlation index, red: high correlation index.

D    Survival analysis of the GBM patients exhibiting high (red) or low (green) IRE1 activity. Student's *t*-test was used with Welch's correction when SD different.

E–G    Expression of microglial/monocyte/macrophage (IBA1, CD14, CD163) (E), angiogenesis (CD31, vWF) (F), and migration/invasion (RHOA, CYR61, CTGF) (G) markers mRNA in the IRE1high (red) and IRE1low (green) populations. Probe analysis was carried out in data from 258 and 265 tumors in IRE1high and IRE1low groups, respectively. Horizontal lines indicate median; box lines indicate first & third quartiles; whiskers indicate min & max. Student's *t*-test was used with Welch's correction when SD different.

H    FACS analysis of CD45/CD11b in freshly dissociated GBM tumors exhibiting high or low IRE1 activity.

I    FACS analysis of CD31 in freshly dissociated GBM tumors exhibiting high or low IRE1 activity. In both cases, 7AAD was used to exclude dead cells.

J    Relative distribution of the different classes of GBM—pro-neural (blue), neural (orange), classic (green), and mesenchymal (red) according to the tumor status, namely IRE1high or IRE1low.

also to the presence of somatic mutations in the IRE1 coding gene. Previous tumor sequencing studies identified IRE1 mutations that were defined as driver in various cancers among which three were found in GBM (Greenman *et al*, 2007; Parsons *et al*, 2008). Here, we sequenced the IRE1 gene (*ERN1*) exons in 23 additional GBM samples and identified a fourth IRE1 mutation in one GBM human sample (Appendix Fig S3A). This somatic A414T mutation came from an aggressive, mesenchymal-like GBM developed in a 70-year-old female. Immunohistochemistry staining revealed that this tumor was also highly vascularized (CD31 staining) and showed strong XBP1s staining (Appendix Fig S3B). Sequence alignment indicates that whereas the mutations P336L, S769F, and Q780* affect conserved amino acids in various species, the mutation identified in our sequencing study altered an apparently less conserved amino acid, which was only conserved in dog, chimpanzee, and human but not in rodents (Appendix Fig S3C). This property could explain why the A414T mutation, previously described in GBM samples, has been excluded from further analyses, as it was considered as a SNP or a secondary acquired mutation (Cancer Genome Atlas Research Network, 2008; Parsons *et al*, 2008). Interestingly since the first discovery of IRE1 somatic mutations in cancers in 2007, a number of cancer exome or whole-genome sequencing studies have also reported around 50 mutations but none of them in GBM (Chevet *et al*, 2015).

### Different kinase and RNase activities of IRE1-related cancer variants

IRE1 is a bifunctional protein that contains a kinase and a RNase domain involved in three downstream signaling pathways including (i) activation of stress pathways [i.e., JNK and NFKB (Hetz, 2012)], (ii) the degradation of targeted RNAs (RIDD), and (iii) the unconventional splicing of XBP1 mRNA. The localization of IRE1 mutations found in cancer revealed no apparent clustering of the mutations in the secondary structure, not even into IRE1 catalytic domains. However, the "cytosolic" mutations S769F and Q780* are located in the kinase domain of the protein whereas the "luminal" mutations P336L and A414T are located in putative alpha-helical domains (Appendix Fig S3D). To measure the potential impact of the four mutations found in GBM, we overexpressed either the wild-type (WT) or the mutated forms of IRE1 in U87 cells, in a normal endogenous IRE1 background (Appendix Fig S4A). The four variants were overexpressed in U87 cells using a lentivirus system, and as anticipated, the stop mutation Q780* leads to overexpression of a shorter IRE1 protein (80 kDa instead of 110 kDa). Finally, immunofluorescence studies showed that IRE1 staining co-localized with an ER marker (KDEL staining) and thereby confirmed that mutations did not affect IRE1 localization to the ER (Appendix Fig S4A and B).

As reported in other cellular system (Han *et al*, 2009), the overexpression of the WT form in U87 was sufficient to activate IRE1 in basal conditions compared to the control empty vector (EV)-expressing cells, as indicated by basal IRE1 phosphorylation, as well as XBP1 mRNA splicing (Fig 2A and B). As expected, Q780* corresponded to a loss-of-function mutation regarding the splicing of XBP1 mRNA. Indeed, the loss of the last fragment of the kinase domain and the entire RNase and C-terminus domains did not affect IRE1 oligomerization but impaired the resulting trans-autophosphorylation (Fig 2B and Appendix Fig S4C) as well as XBP1 mRNA splicing (Fig 2B). Expression of the Q780* variant also prevented XBP1 mRNA splicing by endogenous IRE1 in response to tunicamycin treatment (Appendix Fig S4D). In addition, P336L and A414T mutations increased IRE1 oligomerization capacity (Appendix Fig S4C), leading to IRE1 hyperphosphorylation and enhanced XBP1 splicing (Fig 2A and B). It is important to note that WT-IRE1 overexpression efficiently increased RIDD activity on PERIOD1 (PER1), COL6A1, and SCARA3 mRNAs, whereas little impact was observed on other previously reported RIDD substrates such as SPARC, PDGFRbeta, and VEGF-A mRNAs, thereby pointing toward RIDD selectivity associated with IRE1 variants. As such, the four mutations had different effects depending on the targeted mRNA (Fig 2C). This substrate selectivity might result from modifications in IRE1 binding to luminal or cytosolic partners due to IRE1 overexpression or mutations (i.e., altered oligomerization or signaling properties). Finally, following the observation of Upton and colleagues (Upton *et al*, 2012), we found that IRE1 variant RNase activity controlled miR-17 (miR-17-5p) expression in GBM. Indeed, the A414T variant led to increased miR-17 expression under basal conditions while the P336L variant led to low miR-17 levels (Fig 2D). IRE1 RNase inhibition mediated by MKC4485 (Volkmann *et al*, 2011) restored the expression of miR-17 in P336L IRE1 variant expressing U87 cells (Appendix Fig S4E), thus confirming the involvement of IRE1 RNase in miR-17 expression. Tunicamycin-induced ER stress engaged IRE1 activation and led to further miR-17 degradation (Appendix Fig S4F). We have summarized the differential impact of IRE1 variants on distinct downstream signaling outputs in Fig 2E.

### U87 phenotype and signaling upon expression of IRE1 variants

To further investigate the impact of IRE1 variant expression in U87 cells, we first evaluated the cellular phenotypes generated using phase microscope imaging. All the cells presented a mesenchymal phenotype comparable to U87 transfected with an empty vector (Fig 2F) or parental U87 (not shown), except for those expressing the IRE1 P336L variant, which displayed an epithelial-like phenotype. These cells exhibited similar proliferation rates (Appendix Fig S5A) and still had the capacity of forming spheres in culture (Appendix Fig S5B) as described previously (Dejeans *et al*, 2012). To further evaluate the IRE1-dependent signaling aspects in GBM, we used the KEGG pathway for glioma that compiles the main actors involved in gliomagenesis, and identified the components that were previously shown to be directly or indirectly regulated by IRE1. This revealed that 45% of the components comprised in this pathway were controlled through IRE1-dependent mechanisms (Appendix Fig S5C; yellow boxes). We then monitored the expression levels of PDGFRbeta and p53 (respectively top and bottom of the pathway, Appendix Fig S5D). PDGFRbeta expression was highly expressed in cells expressing EV and the P336L mutant. Interestingly, the expression of wild-type p53 in U87 cells (Cerrato *et al*, 2001) was upregulated by 15-fold in P336L-expressing cells. In those cells, p53 mRNA was not altered compared to EV cells (Appendix Fig S5E) and no mutations were found by sequencing (not shown), thereby suggesting a translational regulation of the protein. To further characterize the impact of the IRE1 variants on cell signaling pathways, we used a transcriptomic approach.

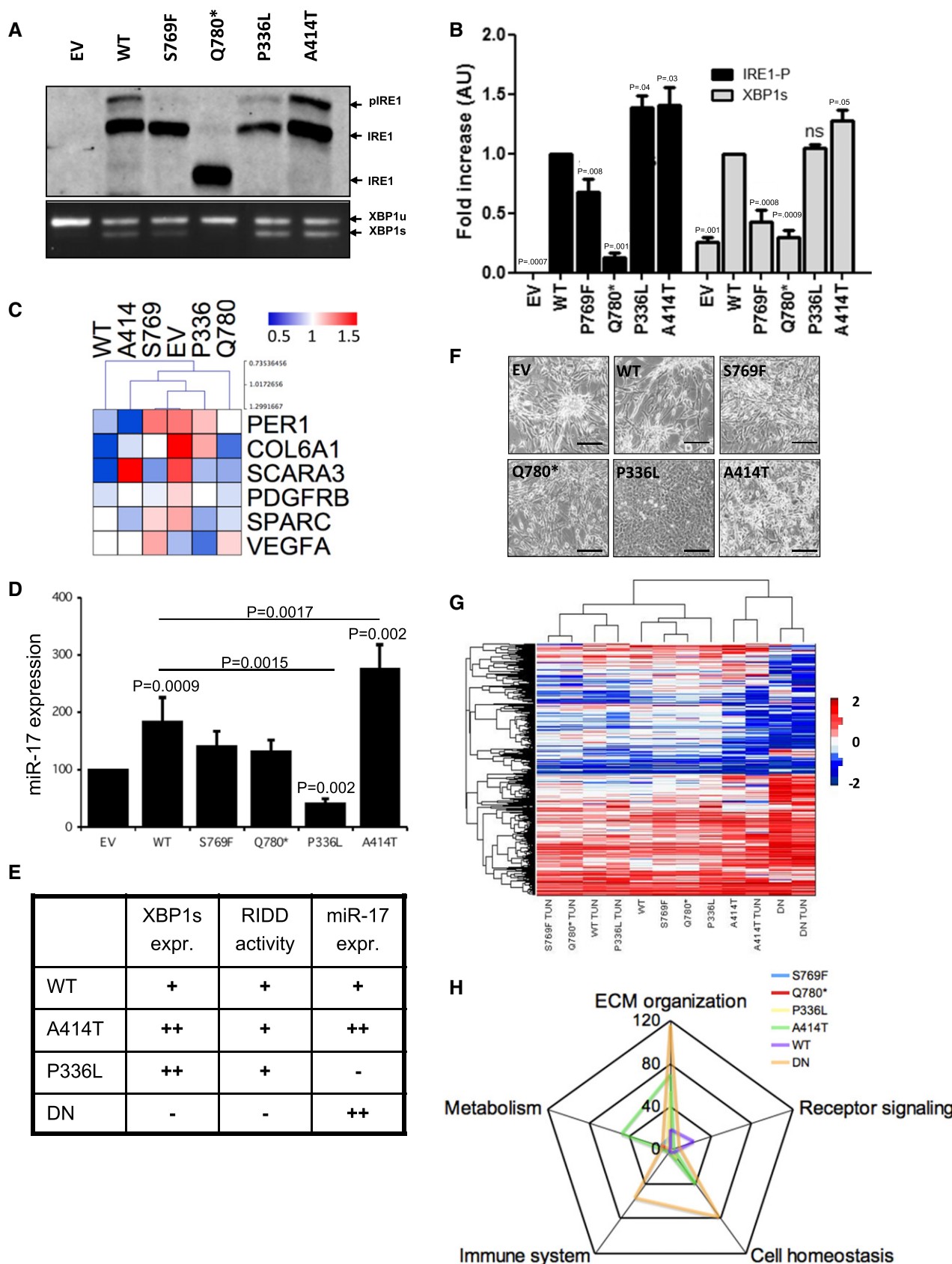

**Figure 2.**

◄

**Figure 2. Impact of somatic mutations on IRE1 signaling.**

A   Anti-IRE1 Phostag immunoblot showing both phosphorylated (p-IRE1) and non-phosphorylated (IRE1) IRE1 proteins revealed IRE1 phosphorylation in basal conditions due to overexpression of WT, P336L and A414T but not S769F nor Q780* forms of IRE1. EtBr-stained agarose gel of XBP1 cDNA amplicons corresponding to unspliced (XBP1u) and spliced (XBP1s) forms of XBP1 mRNA revealed XBP1 splicing in basal conditions due to overexpression of WT, S769F, P336L and A414T but not Q780* forms of IRE1.

B   Bar graph representing the quantification of three levels of IRE1/XBP1 activation: IRE1 phosphorylation (p-IRE1/IRE1) and XBP1 mRNA splicing (XBP1s/(XBP1u + XBP1s)) measured as indicated in (A). Three independent biological samples were used. Data from five independent experiments are presented as means ± SD. P-values are indicated. Student's t-test was used.

C   Heat-map representation of RIDD of mRNA target expression (normalized to 18S) after 2-h Actinomycin D (ActD) treatment. Three independent biological samples were used.

D   Analysis of miR-17-5p expression by RT-qPCR in IRE1 variant expressing cells under basal conditions. Data from three independent experiments are presented as means ± SD. Student's t-test was used.

E   Recapitulative table of the signaling properties of IRE1 WT, DN, P336L, and A414T variants.

F   Phenotypic characterization of U87 cells expressing the different IRE1 variants and imaged by phase contrast microscopy. Scale bar = 100 μm.

G   Heat-map representation of the transcriptomes of U87-expressing variants upon basal conditions or ER stress (induction by tunicamycin).

H   Schematic representation of the signaling pathway enrichment based on the transcriptome data.

Hierarchical clustering revealed that WT IRE1 grossly behaved as the S769F, Q780*, and P336L variants under basal and stress conditions. In contrast, cells expressing the IRE1 A414T variant exhibited a very different gene expression profile than the other cell types that more closely resembled the signature observed in IRE1-DN cells (Pluquet et al, 2013). The expression profiles were then analyzed for signaling pathway activation and unveiled possible pathways selectively activated by IRE1-related cancer variants (Fig 2G). Functional analysis of the gene enrichment pattern indicated a major impact of IRE1 A41AT mutation on signaling pathways involved in metabolism control, extracellular matrix (ECM) organization, and cell homeostasis maintenance, whereas IRE1-DN impacted mostly genes related to ECM organization, cell homeostasis, and the immune system. Interestingly, the impact of other variants on basic cellular signaling functions remained limited compared to the pattern elicited by IRE1 A414T and DN expression (Fig 2H and Appendix Table S1).

**Modulation of tumor development *in vivo* upon expression of IRE1 variants**

To evaluate the significance of each IRE1 variant to tumor growth *in vivo*, we implanted control U87 or cells expressing WT and mutated forms of IRE1 into mouse brain, as previously described (Auf et al, 2010; Pluquet et al, 2013). Fifteen days post-implantation, five animals of each group were sacrificed and brains isolated for immunofluorescence (IF) staining of tumor cells (vimentin) and vessels (CD31). As expected, IRE1 overexpression impacted tumor growth and vascularization, whereas impairment of IRE1 signaling (IRE1-DN) reduced both size and vascularization of the tumors (Fig 3A). An exception of this tumorigenic effect of IRE1 was observed with the P336L mutation. Indeed, injection of U87 expressing the IRE1 P336L never led to the formation of a visible tumor (> 15 injections). This phenomenon may be a consequence of the observed overexpression of the tumor suppressor p53 in those cells (Appendix Fig S5), leading to the attenuation of U87 aggressive phenotype.

Among the four mutations, the loss-of-function mutations S769F and Q780* appeared to have little effects on mouse survival (Fig 3B); however, the Q780* mutation accelerated the early steps of tumor growth compared to the control tumors (orange vs. black lines; Fig 3B). Remarkably, expression of the P336L and A414T

variants, which exhibited similar gain of function on IRE1 *in vitro*, showed diametrically opposed behaviors *in vivo* on tumor development. Indeed, whereas P336L totally blocked tumor formation, A414T shortened mouse survival (Fig 3B), most likely by promoting tumor growth and vascularization with hallmarks of vessel cooption (Fig 3A, bottom). Interestingly, tumors formed from EV, WT, and A414T cells showed high XBP1s expression as assessed by immunohistochemistry which did not account for the differences observed in mouse survival (Fig 3C). Remarkably, the pro-angiogenic effects of the A414T mutation not only increased the number of vessels associated with the tumor mass but also increased the size of those vessels (Fig 3D and E), an effect that was much less visible in early steps of tumorigenesis (Fig 3A and E). Furthermore, the impact of the A414T mutation on the immune infiltrate to the tumor site was also evaluated *in vivo* and showed that expression of this IRE1 variant in U87 cells resulted in the formation of tumors presenting very low levels of macrophage infiltration (F4/80 staining; Fig 3F). This was not the case for other variants (Fig 3G). Finally, tumor-infiltrating spots were quantified as previously described (Pluquet et al, 2013) and showed major infiltration/invasion of DN as previously observed (Drogat et al, 2007; Auf et al, 2010; Jabouille et al, 2015) but also a significant positive impact of the expression of the A414T variant (Fig 3A and H). Thus, our results suggest that selective genetic alterations affecting IRE1 activity condition the specific biological outputs observed at the level of tumor growth, survival, angiogenesis, and immune cell infiltration.

**IRE1 downstream signals drive changes in the tumor microenvironment**

To further dissect the contribution of signals downstream of IRE1 in GBM tumor phenotypes, we took advantage of the properties of IRE1 variants. Global analysis of our results highlights the signaling differences driven by IRE1 mutant forms where WT IRE1, the P336L, and A414T variants exhibited high XBP1 mRNA splicing and RIDD activities whereas the expression of miR-17 was elevated in cells expressing WT or IRE1-DN as well as those expressing the A414T variant (Figs 2E and 4A). Based on this analysis, we reasoned that genes whose expression was upregulated in WT, P336L, and A414T could reflect targets under the control of XBP1s. Using the transcriptome profiles established in Fig 4, we identified a list of 40 genes that segregate with high XBP1s levels (Fig 4B and

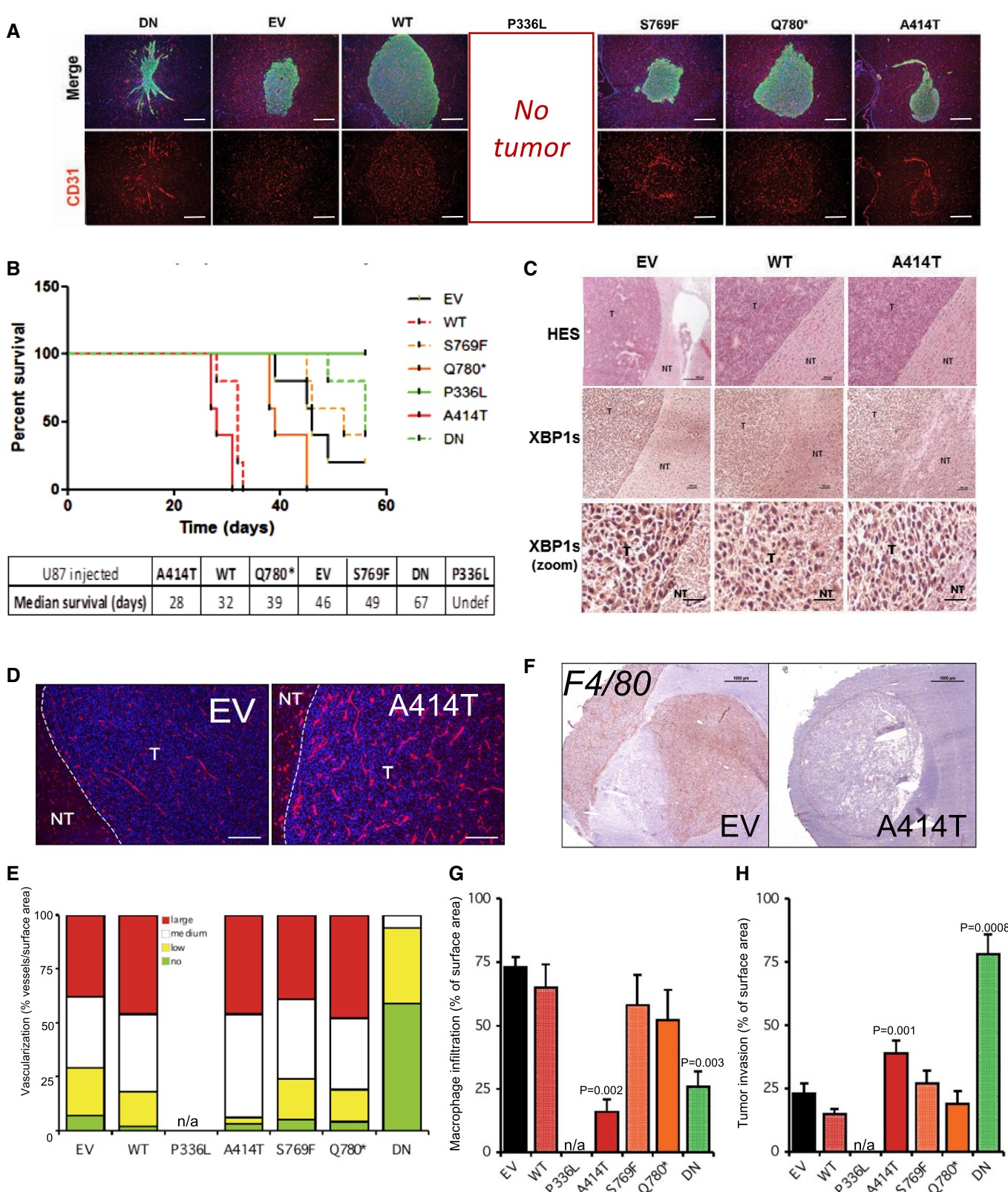

**Figure 3.**

Appendix Table S2). We then used the XBP1s signature to classify tumors from the GBMmark cohort (Appendix Fig S6A) and analyzed the expression of IBA1, CD31, and RHOA in tumors exhibiting high

or low expression of XBP1s target genes (Fig 4C). This revealed that the three markers studied showed higher expression levels in tumors with high XBP1s target gene expression. This was further

**Figure 3.  Impact of IRE1 somatic mutations on tumor development.**

A   Tumor cells (U87) were injected into the brain of recipient mice (5 to 15 Rag-γ 2C$^{-/-}$ per condition). Animals were sacrificed 15 days post-injection. Brains were collected and analyzed by immunofluorescence with anti-vimentin (green) and anti-CD31 (red) antibodies (scale bar = 500 μm).

B   Tumor cells (U87) were injected into the brain of recipient mice (Rag-γ 2C$^{-/-}$). Animals were sacrificed at first clinical sights of tumor development, and each sacrifice was reported in the Kaplan–Meier curve, indicating a gain of lethality for tumors formed in WT or A414T conditions.

C   Brains were collected and analyzed by H&E and immunostaining with anti-XBP1s antibodies (scale bar in top row = 50 μm; scale bar in middle row = 150 μm; scale bar in bottom row = 700 μm). T: tumoral tissue; NT: non-tumoral tissue.

D   CD31 staining (red) and nuclear nucleus staining (blue) in tumors collected at sacrifice exemplified with control (EV) and A414T-expressing cells (scale bar = 250 μm).

E   Graphic representation of tumor vascularization for each tumor. Tumors were classified into four groups relative to their degree of vascularization: (i) avascular tumors (no): no apparent blood vessels in the tumor; (ii) poorly vascularized (low): A few blood vessels are seen in the tumor; (iii) moderately vascularized (medium): Numerous blood vessels are seen in the tumor; (iv) highly vascularized (high): The tumor surface is covered with blood vessels (as well as wide). For animal experimentation, data shown are the mean ± SEM of five tumors per experiment.

F   F4/80 (macrophage) staining in tumors collected at sacrifice exemplified with control (EV) and A414T expressing cells. Scale bar = 100 μm.

G   Quantification of macrophage infiltration in tumors, data are represented as the mean ± SEM of five tumors per experiment. Statistics were determined using two-way ANOVA with Bonferroni post-test.

H   Quantification of tumor invasion as described previously. Tumors were stained with anti-vimentin antibodies, and the number of vimentin-decorated spots was quantified per surface area as previously described (Pluquet et al, 2013). Data are the mean ± SEM of five tumors per experiment. Statistics were determined using two-way ANOVA with Bonferroni post-test.

confirmed using immunohistochemistry with antibodies against XBP1s and IBA1. A total of 35 cases of GBM were analyzed with anti-XBP1s and revealed either no staining (Fig 4D–1) or staining in the nucleus (Fig 4D–2) and in the cytoplasm (Fig 4D–3/4). A subset of those tumors (*n* = 24) was then analyzed for IBA1 expression (Appendix Fig S6B), and a correlation between the presence of IBA1 and that of XBP1s was established thereby indicating that high XBP1s in the tumor may control immune cell (macrophage) infiltration (Fig 4E).

Next, we investigated how RIDD activity could impact on tumor characteristics. To this end, we determined a potential RIDD signature based on the ability of IRE1 to cleave select mRNA *in vitro*. This screening identified a group of 1,141 mRNAs susceptible to be cleaved *in vitro* by IRE1 (Appendix Fig S7A and Table S3), which were then intersected with the set of genes upregulated in IRE1-DN cells. This analysis yielded a subset of 37 potential GBM-specific RIDD targets (Fig 4F and Appendix Table S4). Their functional annotation suggests the enrichment in genes involved in the NOD pathway, interaction with the environment, and biogenesis of cellular components (Appendix Fig S7B). Then, this cluster of mRNAs was used to identify RIDD-positive and RIDD-negative tumor populations in the GBMmark cohort (Fig 4G). The expression of immune infiltration, angiogenesis, and invasion markers in these populations confirmed previous results and ruled out tumoral RIDD of mRNA in the recruitment of immune cells (Fig 4H). In summary, in contrast to the IRE1/XBP1 axis that exhibits pro-tumorigenic signaling features, the RIDD of mRNA pathway may antagonize tumor invasion and angiogenesis with no significant effect on immune cells infiltration.

**Differential contribution of RIDD and XBP1 mRNA splicing to GBM**

We then investigated the role of IRE1/miR-17 axis in cancer progression. To this end, we took advantage of the properties of the different IRE1 variants toward miR-17 (Fig 4A) and established a minimal group of genes whose expression could be under the control of miR-17 (Fig 5A and Appendix Table S5). This set of genes is involved in morphogenetic programs, cell adhesion, synthesis of aromatic compounds, and to a lesser extent in the response to reactive oxygen species (Fig 5B). This information was then used to

evaluate tumors with high or low IRE1/miR-17 in the GBMmark cohort and to monitor the expression of IBA1, CD14, CD31, vWF, RHOA, and CTGF (Fig 5C). As for RIDD of mRNA, these data indicated that RIDD for miR-17 exhibited anti-angiogenic and anti-migratory effects. This led us to correlate high RIDD IRE1 activity, which might lead to low miR-17 expression, and better outcome in GBM patients. To test this hypothesis, we evaluated the expression of miR-17-5p in 30 GBM tumors and identified two groups of tumors exhibiting low or high miR-17-5p expression (Fig 5D). Patient survival was evaluated in those two groups of patients and revealed that low miR-17-5p levels in tumors correlate with better survival than those patients presenting high miR-17-5p tumors (Fig 5E), thereby confirming our initial hypothesis. To functionally explore the role of the IRE1/miR-17 axis in GBM development, we blunted miR-17 activity with anti-miR-17 sponges in EV and A414T cells and tested the expression of predicted miR-17 target genes. We confirmed that antagonizing miR-17 increases the expression of miR-17 targets in cells expressing the A414T IRE1 variant (Fig 5F). To further evaluate the functional role of miR-17 in GBM, we monitored the impact of either XBP1 silencing or antagonizing miR-17 in U87 or U251 cells expressing an empty vector or the IRE1 A414T variant. This approach revealed that siRNA-mediated XBP1 silencing in U87 cells impaired monocyte chemoattraction (Fig 5G) whereas antagonizing miR-17 did not have any significant effect (not shown). In addition, we found that overexpression of IRE1 A414T in U251 cells increased cell migration using a trans-well assay, a feature that was impaired by XBP1 silencing or miR-17 buffering (Fig 5H).

Our data led us to propose a complex model in which the IRE1/XBP1 signaling axis would promote GBM aggressiveness through the enhancement of tumor immune infiltration and angiogenesis as well as tumor cell invasiveness properties, whereas RIDD (of mRNA and miRNA) would play an anti-tumoral role by selectively reducing tumor angiogenesis as well as tumor cell invasiveness (Fig 6A). To further demonstrate the divergent activities of IRE1 in cancer, the TCGA cohorts (microarrays and RNAseq) were analyzed for populations exhibiting low and high XBP1 splicing and RIDD activities. Hierarchical clustering revealed four major groups as follows XBP1s$^{high}$/RIDD$^{low}$  (XBP1$^+$/RIDD$^-$);  XBP1s$^{low}$/RIDD$^{low}$  (XBP1$^-$/RIDD$^-$);  XBP1s$^{low}$/RIDD$^{high}$  (XBP1$^-$/RIDD$^+$);  and  XBP1s$^{high}$/RIDD$^{high}$  (XBP1$^+$/RIDD$^+$) (Fig 6B and C, see Materials and

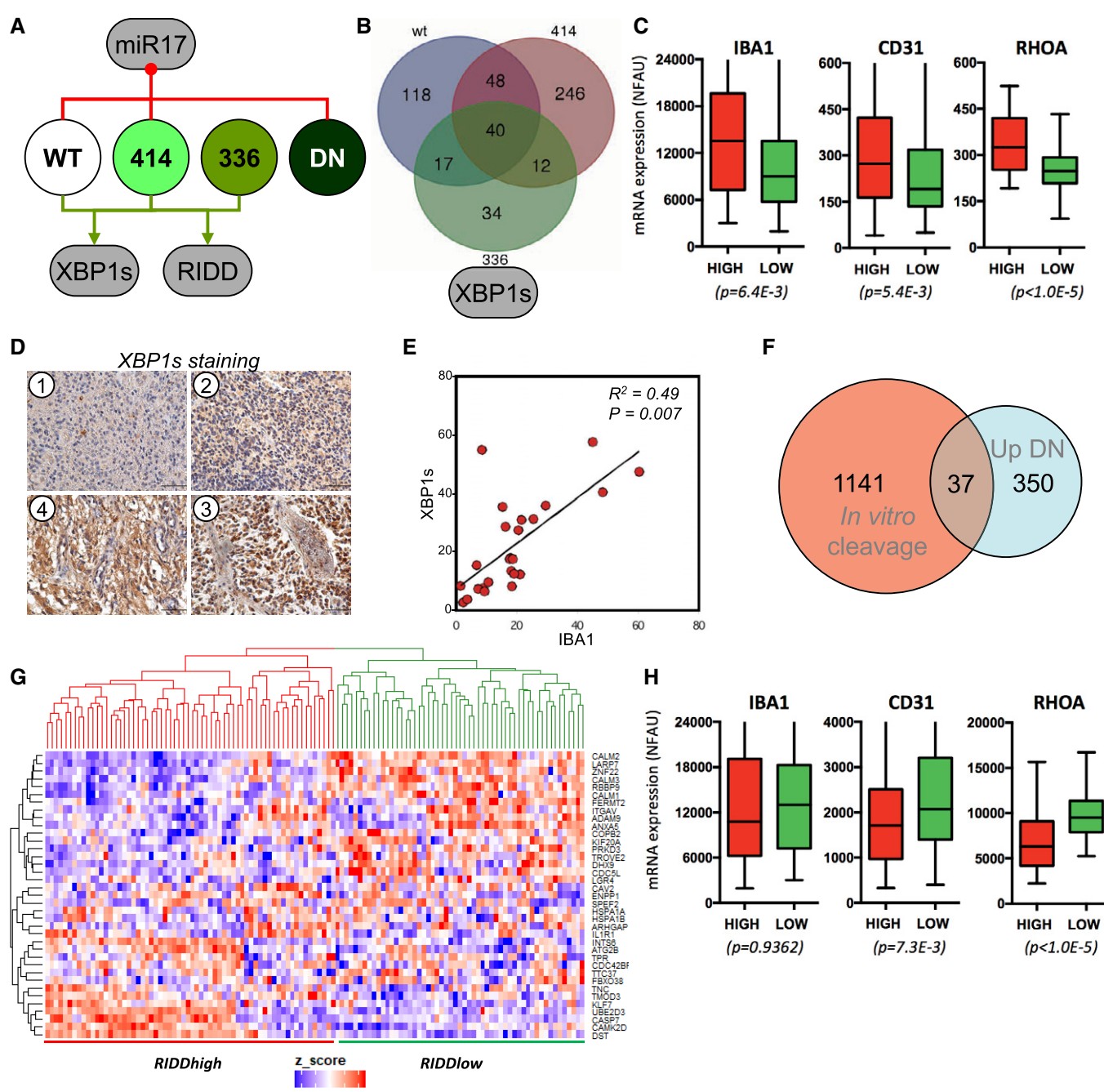

**Figure 4. XBP1s and RIDD of mRNA signals in GBM.**

A   Schematic representation of the signaling characteristics of IRE1 variants based on data generated in Fig 2, toward XBP1 mRNA splicing, RIDD of mRNA, and RIDD of miR-17.

B   Intersection of the upregulated genes in WT, P336L, and A414T-expressing cells as the hallmark of XBP1s expression. A list of 40 genes was established and confronted to the tumor transcriptomes of the GBMmark cohort (Appendix Table S2). XBP1s high and XBP1s low groups of tumors were established.

C   Expression analysis of IBA1, CD31, and RHOA mRNA in the XBP1s high (red; *n* = 44) and low (green; *n* = 75) groups of tumors. Horizontal lines indicate median; box lines indicate first & third quartiles; whiskers indicate min & max. Student's *t*-test was used with Welch's correction when SD different.

D   Characterization of XBP1s high and low tumors using immunohistochemistry (no XBP1s expression (1), and increasing amounts of XBP1s expression (2–4) are shown).

E   Correlation of XBP1s and IBA1 staining.

F   Intersection of the list of RNA cleaved by IRE1 *in vitro* (Appendix Table S3) and that of mRNA whose expression is upregulated in IRE1-DN cells under basal conditions.

G   A list of 37 genes was established and confronted to the tumor transcriptomes of the GBMmark cohort. RIDD[high] and RIDD[low] groups of tumors were established.

H   Expression analysis of IBA1, CD31, and RHOA mRNA was evaluated in the RIDD[high] (red; *n* = 64) and RIDD[low] (green; *n* = 55) groups of tumors (I). Horizontal lines indicate median; box lines indicate first & third quartiles; whiskers indicate min & max. Student's *t*-test was used with Welch's correction when SD different.

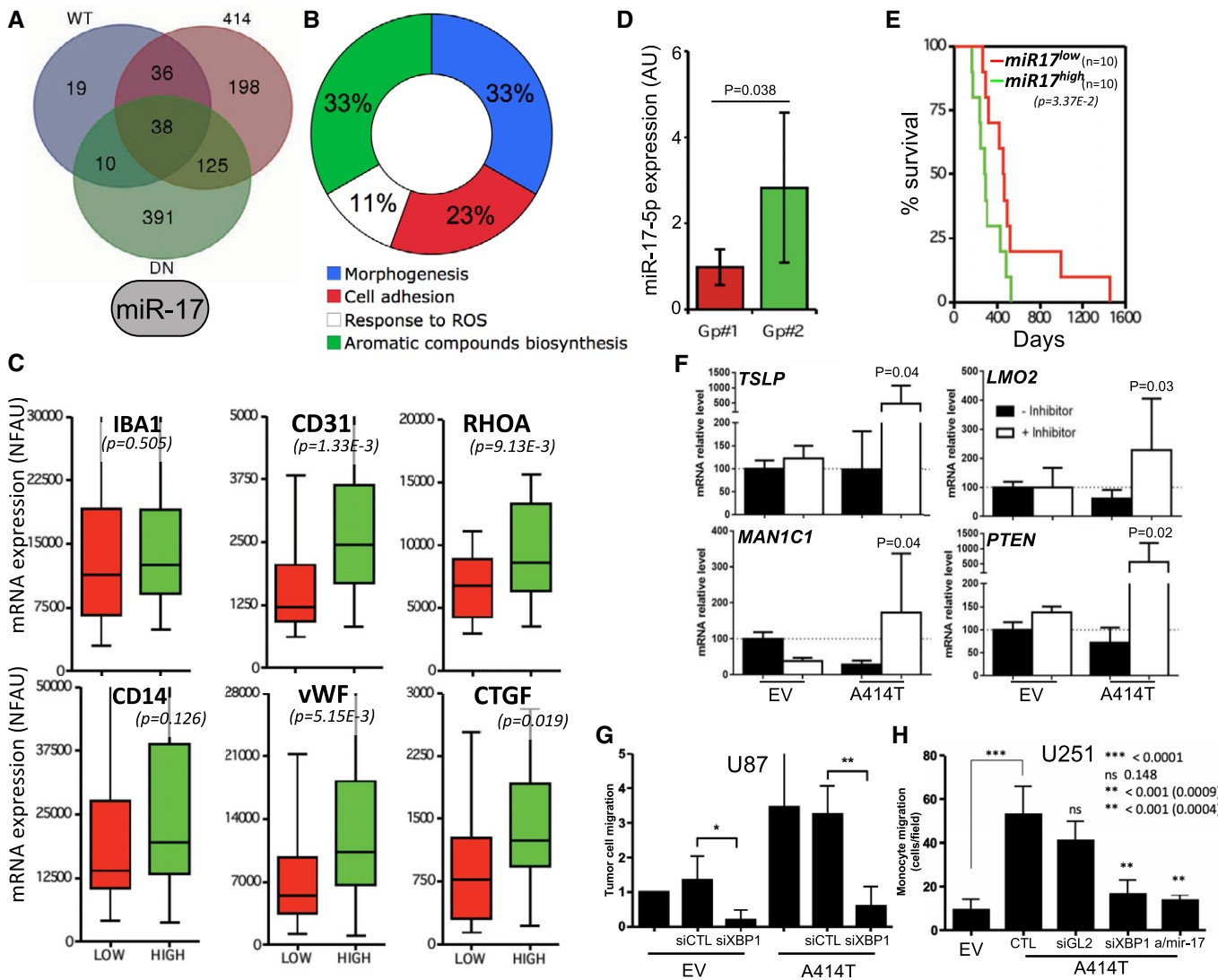

**Figure 5. RIDD of miRNA (miRIDD) signals in GBM.**

A    Intersection of the downregulated genes in WT, DN, and A414T-expressing cells (high miR-17) yielding a list of 38 genes.

B    These 38 genes were analyzed for being predicted direct miR-17 targets, which reduced the list to 12 genes (Appendix Table S5). Their functional annotation is shown.

C    The 12 genes list was confronted to the tumor transcriptomes of the GBMmark cohort. miRIDD[high] and miRIDD[low] groups of tumors were established, and expression of IBA1, CD14, CD31, vWF, and RHOA, CTGF mRNA was evaluated in the miRIDD[high] (green; $n = 10$) and miRIDD[low] (red; $n = 18$) groups of tumors. Horizontal lines indicate median; box lines indicate first & third quartiles; whiskers indicate min & max. Student's $t$-test was used.

D    Expression of miR-17-5p in 30 GBM tumors and distribution of these tumors in low (red) and high (green) miR-17-5p groups. Student's $t$-test was used.

E    Kaplan–Meier survival curves of low (red) and high (green) miR-17-5p GBM tumor patients. Student's $t$-test was used.

F    Real-time PCR analysis of three miR-17 targets identified in our study (TSLP, LMO2, MAN1C1) and a positive control (PTEN) on mRNA extracted from control or IRE1 A414T expressing U87 cells. Student's $t$-test was used.

G    Monocyte chemoattraction with medium conditioned by U87 cells expressing an empty vector or the IRE1 A414T variant and silenced or not for XBP1. Student's $t$-test was used. *$P$-value = 0.0015, **$P$-value = 0.0009.

H    Glioblastoma cell (U251) migration assay in Boyden chamber carried out with cells expressing or not IRE1 A414T and either silenced for XBP1 or transfected with an antago-miR-17. Student's $t$-test was used.

Data information: Data from five independent experiments are presented as means ± SD.

Methods). Remarkably, patient survival analysis in the XBP1[−]/RIDD[+] and XBP1[+]/RIDD[−] populations revealed a clear segregation where the former group survived statistically longer than the latter as suggested from our working model (Fig 6D and E). Moreover, as anticipated, XBP1[+]/RIDD[−] GBM exhibited higher expression IBA1, CD31, and RHOA than the XBP1[−]/RIDD[+] tumors (Fig 6F and G).

This was also confirmed by the fact that the XBP1[+]/RIDD[−] population was enriched in mesenchymal tumors whereas XBP1[−]/RIDD[+] comprised more pro-neural and neural tumors (Appendix Fig S7C). Interestingly, when the splicing of XBP1 was evaluated in the TCGA RNAseq cohort and compared to the XBP1[+] groups established using our method, we correlated both information thereby

confirming the validity of our approach (Appendix Fig S7D). Finally, differences in the survival of the patients belonging to the intermediate groups (i.e., XBP1$^-$/RIDD$^-$; XBP1$^+$/RIDD$^+$) were not statistically significant (Appendix Fig S7E). Taken together, these experiments suggest an antagonistic role of XBP1 mRNA splicing and RIDD in GBM specifically regarding cell migration and angiogenesis.

### Modeling IRE1 contribution to GBM development in primary GBM lines

We then tested whether the tumor classification established in Fig 6 was also relevant in primary GBM lines that could in turn serve as an *in vitro* model for better understanding the role of IRE1 signaling in GBM development. We therefore applied the same clustering method as in Fig 6 on the transcriptome datasets from 12 primary GBM lines. This revealed that the 12 lines clustered into the same four groups as observed for the tumors (Fig 7A). These tumor lines exhibited various phenotypes in culture, notably regarding adhesion/protrusion/migration (Fig 7B). Interestingly, high adhesion/migration also correlated with the XBP1$^+$ status. This was confirmed using RT-qPCR with lines belonging to the XBP1$^+$ group exhibiting the highest XBP1s levels (Fig 7C). This information however was not further correlated with the *in vitro* migration properties of the primary lines (Fig 7D). To further evaluate the relevance of the classification, the 12 lines were orthotopically injected in mice and the resulting tumors evaluated using vimentin staining (Fig 7E). Together with the tumor size (Fig 7F) and the survival data (Fig 7G), these experiments confirmed that XBP1$^+$/RIDD$^-$ tumor cells yielded the most aggressive tumors whereas injecting XBP1$^-$/RIDD$^+$ tumor cells resulted in very small tumors, thereby confirming the results observed in patients' tumors (Fig 6). We then monitored both macrophage recruitment to the tumors as well as angiogenesis and showed that although high XBP1s correlated with important macrophage infiltration in the tumors (Fig 7H), tumoral large vessel content did not significantly change in the different groups (Fig 7I). Interestingly, the expression levels of CCL2 correlated with high XBP1s (Fig 7J) whereas that of VEGF did not (Fig 7K). These data show that primary GBM lines recapitulate, at least partially, the IRE1 signaling properties observed in human tumors and maintain the expected biological outputs even *in vivo*.

We then further tested whether altering IRE1 activity in those cells could impact on their tumoral properties. To this end, we overexpressed IRE1 WT and the Q780* mutant [known to impair XBP1s (Fig 2B)] in four primary lines, namely RNS85, RNS87, RNS96, and RNS130. The lines were selected on the basis of IRE1 mRNA expression (Appendix Fig S8A) and then analyzed by Western blot with anti IRE1 antibodies (Fig 8A). As expected, IRE1 expression was higher upon overexpression of IRE1 WT and the shorter form of IRE1 observed upon expression of the Q780* variant. These different lines were then monitored for the splicing of XBP1 mRNA (Fig 8B) and for the expression of UPR target genes (Fig 8C). Again, these analyses confirmed that in RNS85, 87, 96, and 130, overexpression of IRE1 increased IRE1 activity, thereby resulting in an increased expression of select target genes whereas the expression of the Q780* variant resulted in blunting XBP1s signaling. To further investigate the role of IRE1 modulation in those lines, we first tested how IRE1 activation or inhibition affected the expression of genes related to cell migration (EMT) and chemokine production. This revealed that overexpression of WT IRE1 increased the expression of the EMT-related genes VIM, ZEB1, and TGFB2 whereas that of Q780* IRE1 did not affect it when compared to the control parental line (Fig 8D). Similar results were also observed for the expression of the chemokines CXCL2, CCL2, and IL6 (Fig 8E), thereby confirming the contribution of IRE1 signaling to those pathways.

We then monitored the functional effects of IRE1 modulation in primary lines on tumor cell migration since the cellular phenotypes were altered when expressing IRE1 WT or mutant (Appendix Fig S8B). This was carried out using Boyden chamber-based assays (Fig 8F), and quantitation showed that in most cases IRE1 activation promoted cell migration whereas IRE1 inhibition completely abrogated it in all primary lines tested (Fig 8G). As we previously showed that IRE1/XBP1 signaling was also involved in monocytes chemoattraction, we tested whether modulating IRE1 activity in primary lines would affect this property using Boyden chamber-based migration and FACS analyses (Fig 8H). Quantitation of the results showed that, as observed for migration, enhancing IRE1 activity led to increased chemoattraction whereas blunting XBP1s through the expression of IRE1 Q780* resulted in decreased capacity to attract monocytes in the four primary lines tested (Fig 8I). These results obtained in primary GBM lines confirm our previous observations in U87 cells and in human tumor samples and support our model in which IRE1 activity could control the specific properties of GBM tumor cells through the combined action of XBP1s and RIDD.

## Discussion

Our work demonstrates that in GBM IRE1, downstream signals, including XBP1 mRNA splicing and RIDD, dictate tumor phenotypes and patient outcomes. At first, we showed the relevance of IRE1 signaling in GBM from two independent cohorts (TCGA and GBMmark) and found that high IRE1 activity correlates with shorter patient survival and increased tumor infiltration by immune cells, increased tumor angiogenesis, and enhanced invasion/migration properties of the tumor cells (Fig 1). Previous studies demonstrated the importance of IRE1 signaling for tumor aggressiveness; however, they did not provide any information on the underlying molecular mechanisms involved in this phenomenon. Previous studies established the IRE1 molecular signature using various exogenous acute stresses, such as hypoxia and nutrient deprivation (Pluquet *et al*, 2013). However, the use of such micro-environmental challenges did not recapitulate the complexity of brain cancer as an experimental mean to define the molecular mechanisms downstream of IRE1 involved in tumor growth. Thus, we reasoned that mutations identified on IRE1 in GBM could serve as interesting tools to characterize how specific IRE1-dependent signaling pathways could control tumor phenotypes at both the tumor cell and stroma levels. IRE1 mutations in GBM were previously reported (Chevet *et al*, 2015), but the functional consequences of those mutations on IRE1 signaling remained undocumented. To further expand the repertoire of IRE1 mutations in GBM, we sequenced the IRE1 gene in 23 additional GBM tumors (Appendix Fig S3), and then analyzed the signaling characteristics of the variants identified.

In the present study, we have characterized in detail IRE1 somatic variants using complementary approaches and uncovered a

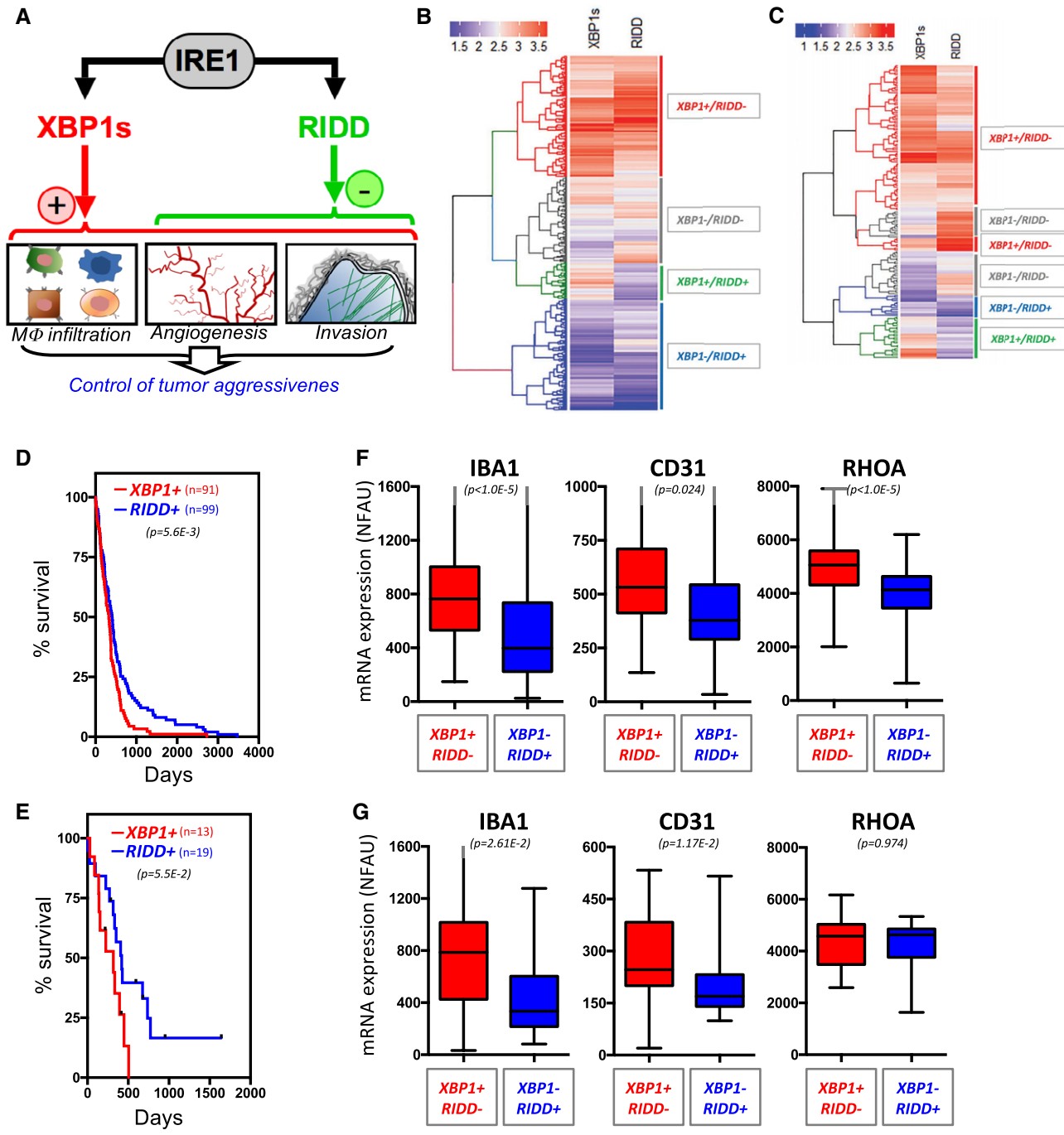

**Figure 6. Deconvolution of IRE1 signaling in GBM tumors.**

A    Schematic representation of the antagonistic XBP1s and RIDD signals in GBM tumors and their biological impact on tumor aggressiveness.

B, C    Hierarchical clustering of GBM patients (TCGA cohort—microarray, B; TCGA cohort—RNAseq, C) based on XBP1s and RIDD scores (see Materials and Methods).

D, E    Kaplan–Meier survival curves of XBP1[+]/RIDD[−] (red) and XBP1[−]/RIDD[+] (blue) GBM patients of the TCGA microarray cohort (D), TCGA RNAseq cohort (E). Student's *t*-test was used.

F, G    Expression of monocyte (IBA1), angiogenesis (CD31), and migration/invasion (RHOA) markers mRNA in the four groups established in the two cohorts. Horizontal lines indicate median; box lines indicate first & third quartiles; whiskers indicate min & max. Student's *t*-test was used.

novel mutation in IRE1 on the A414 residue. Expression of this variant in U87 cells led to highly aggressive cancer with enhanced vascularization and reduced infiltration of macrophages to the orthotopic tumors (Figs 3 and 7). This result was difficult to explain only on the basis of the signaling characteristics of this mutant (Fig 2) and suggested a highly complex integrated IRE1 signal

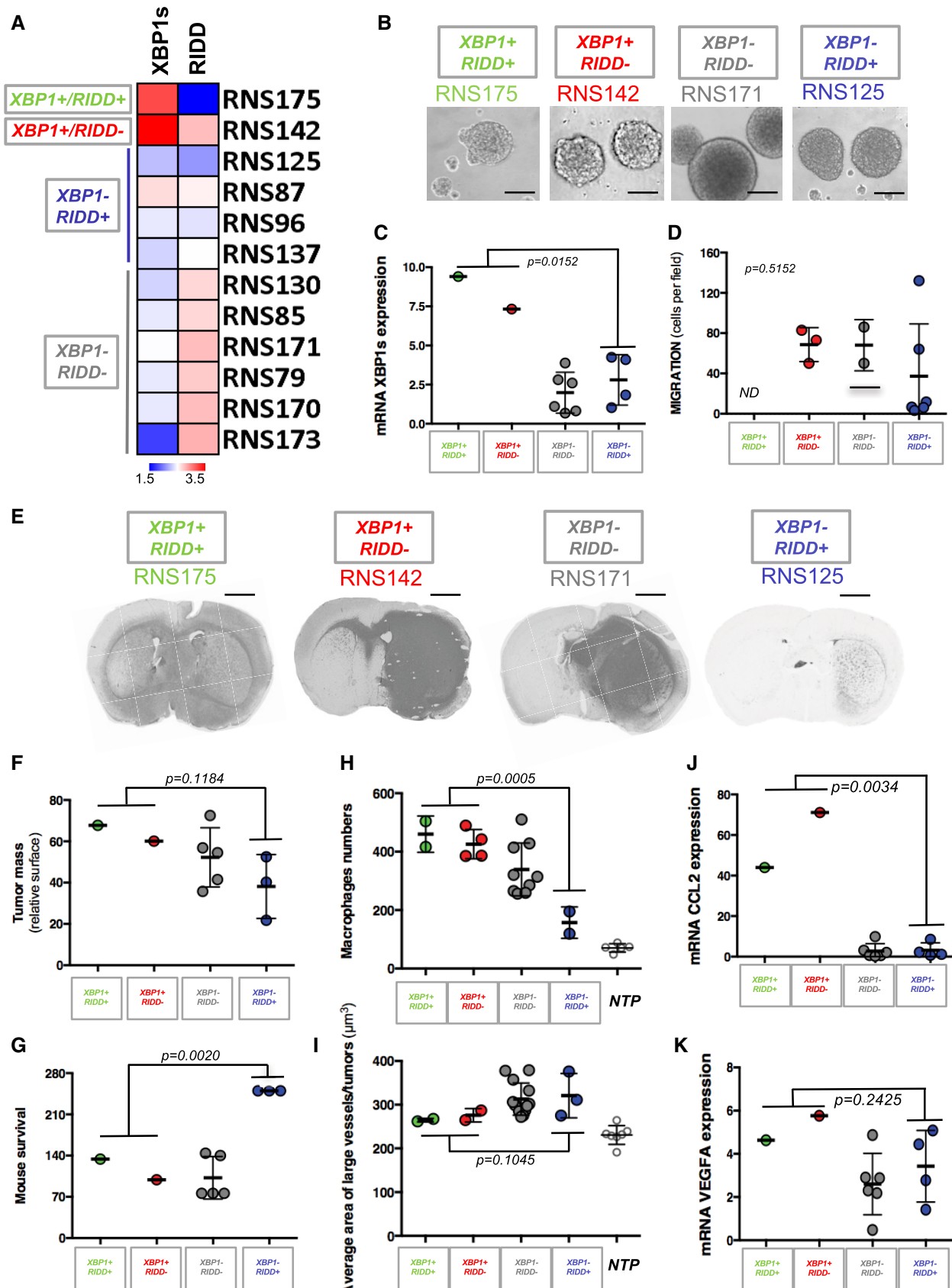

**Figure 7.**

**Figure 7.   Primary GBM lines exhibit IRE1 signaling properties of the parental tumors.**

A    Hierarchical clustering of 12 GBM primary lines based on XBP1s and RIDD scores (see Materials and Methods).

B    Phase contrast images of the phenotypes exhibited by the primary lines from the four groups established in (A) (scale bar = 100 μm).

C    Quantitation of XBP1s mRNA in all primary lines relative to the group to which they belong ($n = 3$, mean $\pm$ SD). ANOVA was used for statistical analyses.

D    Quantitation of the Boyden chamber migration assays for all the lines ($n = 3$, mean $\pm$ SD). ANOVA was used for statistical analyses.

E    GBM primary cell lines were implanted in nude mice. Animals were sacrificed when first clinical signs appeared. Brains were then collected and analyzed for vimentin expression by immunohistochemistry (scale bar = 800 μm).

F    Quantitation of tumor size. ANOVA was used for statistical analyses. For each cell line tested (represented by a single point on the graph), the average of three independent experiments is shown $\pm$SD.

G    Quantitation of mouse survival. ANOVA was used for statistical analyses. For each cell line tested (represented by a single point on the graph), the average of three independent experiments is shown $\pm$SD.

H    Quantitation of macrophage infiltration in orthotopic tumors (IBA1 staining). ANOVA was used for statistical analyses. For each cell line tested (represented by a single point on the graph), the average of three independent experiments is shown $\pm$SD.

I    Quantitation of angiogenesis in orthotopic tumors (CD31 staining). ANOVA was used for statistical analyses. For each cell line tested (represented by a single point on the graph), the average of three independent experiments is shown $\pm$SD.

J, K   Expression of CCL2 (J) and VEGF-A (K) mRNA as determined using RT-qPCR in the different primary GBM lines. ANOVA was used for statistical analyses. For each cell line tested (represented by a single point on the graph), the average of three independent experiments is shown $\pm$SD.

accounting for both XBP1 mRNA splicing and RIDD characteristics to produce the observed tumor phenotype. This was reinforced by our work on the P336L mutation which is, so far, the only IRE1 mutation identified in more than one tumor sample and even in more than one cancer type [one in glioma (Parsons *et al*, 2008) and two in intestinal cancers (Cosmic)], thereby confirming its relationship with cancer development. We thus hypothesized that the oncogenic potential of this mutation may need a particular cancer context, such as the presence of acquired mutation in key genes for GBM development (EGFR, PTEN, TP53, NF1, and IDH1), as no previous study defined P336L as a driver mutation. We showed that TP53 was overexpressed in IRE1-P336L expressing U87 cells which most likely promoted tumor suppression *in vivo* (TP53 wild type in U87; Appendix Fig S5). This observation rules out the driver role of this mutation that would only subsist in a P53 mutant background. Moreover, recent work reported a direct role of the IRE1 target JNK in stabilizing EGFR ligand epiregulin (EREG) and consequently an autocrine activation loop of EGFR, which should provide proliferative advantage of GBM cells in which EGFR signaling was already altered by mutations (Auf *et al*, 2013). This hypothesis could also explain the proliferative effects of A414T mutation, and future studies should define the involvement of EGFR or other key GBM proteins in IRE1-dependent GBM growth, as both P336L and A414T mutations seemed to stabilize IRE1 kinase and RNases activities. Hence, the signaling characteristics of IRE1 variants confer GBM tumors with specificities that could lead to aggressive features.

We took advantage of the signaling characteristics of IRE1 variants associated with GBM and the fact that we characterized the transcriptome of U87 cells expressing those variants under basal and ER stress conditions to delineate specific molecular signatures

of each distinct IRE1 signaling output. This analysis revealed that the XBP1s signaling axis promoted tumor infiltration by immune cells, increased angiogenesis, and enhanced the expression of migration/invasion markers. Interestingly, these properties were also confirmed using approaches relying on immunohistochemical analyses on a different subset of tumors (Fig 4). In contrast, RIDD activity (toward either mRNA or miR-17) attenuated both the angiogenic response and the migration/invasion properties of the tumor cells (Figs 4 and 5). The opposite signals elicited by XBP1s and RIDD confer specific aggressive features to tumors with XBP1$^+$/RIDD$^-$ associated with a worse prognostic outcome than those with XBP1$^-$/RIDD$^+$ properties (Fig 7). Interestingly, tumors with low RIDD/XBP1s and those with high XBP1s/RIDD did not yield different prognoses in terms of patient survival (Appendix Fig S6) most likely due either to the major contribution of other pathways still to be identified (i.e., EGFR, P53) or to compensatory mechanisms of both pathways, respectively. Beyond the characterization of these signaling properties in U87 cells and in human tumors, we also demonstrated that those pathways were also conserved in primary GBM lines and that their modulation had a significant impact on the tumor cells phenotypes (Figs 7 and 8). These pathology-relevant tools will now be useful to define how IRE1 signals toward XBP1s or RIDD are regulated at the level of IRE1 and how those two signaling arms quantitatively interact to drive specific tumor characteristics. Moreover, RIDD of miRNA (miRIDD) activity could interact with known pro-tumoral pathways in GBM through regulation of other IRE1 miRNA targets such as miR-34a (Genovese *et al*, 2012).

In addition to providing the first evidence of the co-existence of antagonistic IRE1 downstream signals in GBM and correlating those with tumor aggressiveness features, our work highlights the

**Figure 8.   Modulating IRE1 activity in primary GBM lines and phenotypic outcomes.**

A    Western blot analysis of the expression of WT or Q780* IRE1 in RNS85, RNS87, RNS96, and RNS130 primary GBM lines.

B    Analysis of XBP1 mRNA splicing using RT-PCR.

C    Quantitation of the expression of CHOP, HERPUD1, ERDJ4, GADD34 (UPR genes) using RT-qPCR in the four lines.

D    Quantitation of the expression of vimentin (VIM), ZEB1, TGFB2 (EMT genes) using RT-qPCR in these IRE1-modified lines.

E    Quantitation of the expression of CXCL2, CCL2, IL-6 (chemokine genes) using RT-qPCR in these modified lines.

F    Representative phase contrast images of migrating RNS85-derived lines tested in the Boyden chamber migration assay (scale bar = 10 μm).

G    Quantitation of tumor migration for all the lines ($n = 3$, mean $\pm$ SD). Student's $t$-test was used for statistical analyses.

H    Representative flow cytometry dot plot graphs obtained in the monocytes chemoattraction assay using RNS85-derived lines conditioned media.

I    Quantitation of the monocytes chemoattraction for all the lines ($n = 3$, mean $\pm$ SD). Student's $t$-test was used for statistical analyses.

Data information: Numbers shown on the top of the graphs presented in (C, D, E) indicate $P$-values, Student's $t$-test was used.

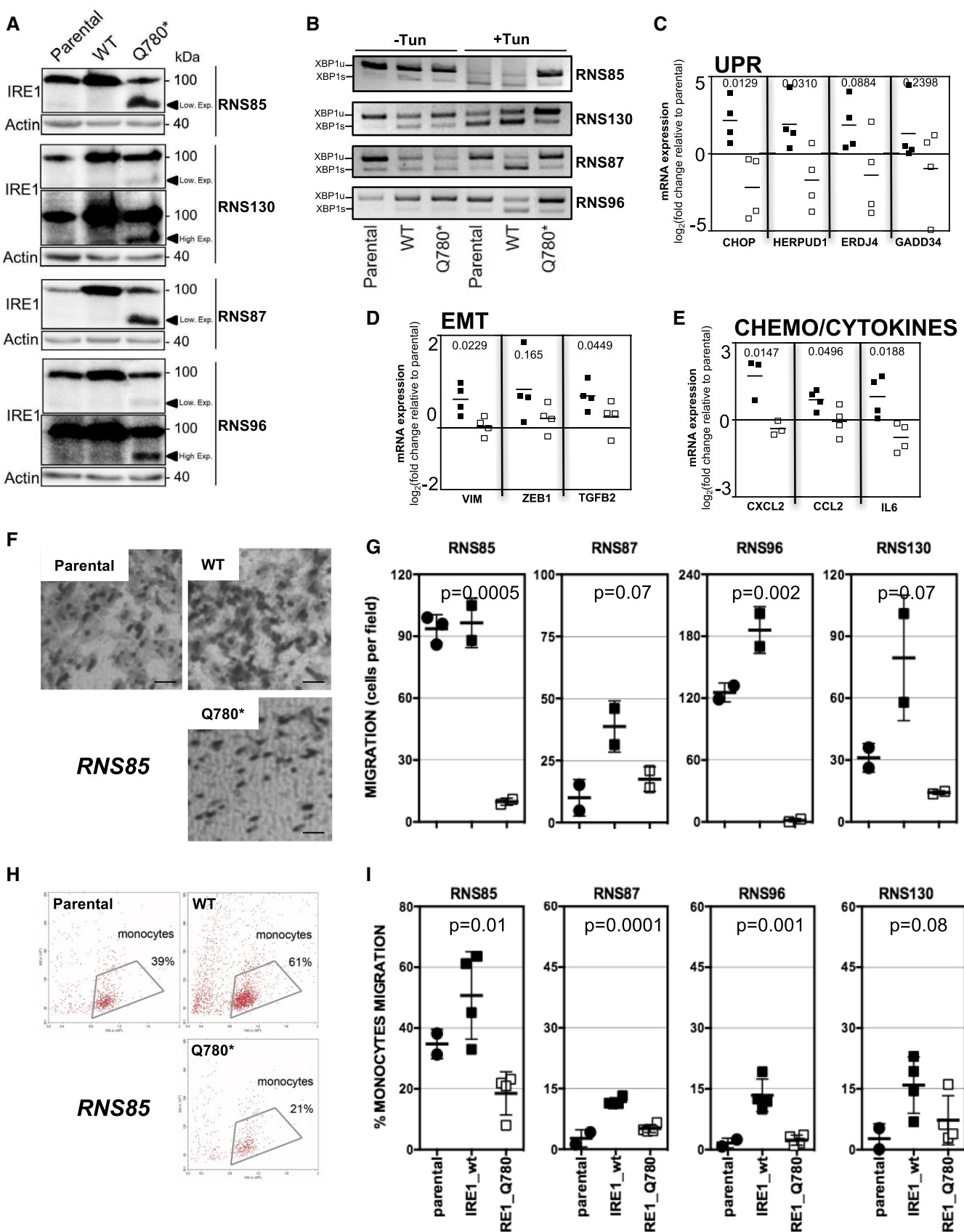

Figure 8.

possibility of using IRE1-targeted therapeutics in cancer. Indeed, it is likely that in XBP1$^+$/RIDD$^-$ tumors, inhibition of IRE1 RNase with small molecules (Hetz et al, 2013) or selective inhibition of the XBP1 mRNA ligase RtcB (Jurkin et al, 2014; Kosmaczewski et al, 2014; Lu et al, 2014; Ray et al, 2014) might lead to significant impairment of tumor growth. This type of strategy would be highly relevant for tumor cells exhibiting constitutive expression of XBP1s such as triple-negative breast cancers (Chen et al, 2014). Similarly, an approach aiming at increasing RIDD activity in XBP1$^-$/RIDD$^+$ tumors to induce cell death mechanisms would be predicted to sensitize GBM cells to chemotherapies such as TMZ. This could for instance be achieved by using inhibitors of BiP that were successfully tested in melanoma models (Cerezo et al, 2016). In addition to the direct effect of IRE1 inhibitors on the tumor cells, one might also consider their use in combination with current therapies, the most common of which comprising the combination of radio- and chemotherapy (the latter with the alkylating agent TMZ; Stupp et al, 2005). Provided that about half of GBM patients are resistant to TMZ, their stratification in terms of TMZ sensitization through selective IRE1 inhibition would represent an appealing therapeutic alternative. Finally, the link between IRE1 signaling and the most recent classification of GBM (Louis et al, 2016) as well as whether or not IRE1 downstream signals are associated with the classes of GBM with poorer (mesenchymal) or better (pro-neural) prognosis (Li et al, 2015) remains to be established.

Collectively, our work demonstrates for the first time that the uncoupling of XBP1s and RIDD signals downstream of IRE1 impacts on cancer development and points toward an alternative therapeutic avenue coupled with personalized molecular diagnosis for (i) decreasing tumor cells' adaptive properties, (ii) enhancing RNA catabolic pathways leading to accelerated tumor cell death, and (iii) modulating the tumor stroma through reduced angiogenesis and increased anti-tumor immunity. These approaches combined with a better knowledge of GBM IRE1 signaling characteristics may contribute to develop a new precision medicine tool for GBM treatment.

# Materials and Methods

### Patient samples and primary lines

The experiments conformed to the principles set out in the WMA Declaration of Helsinki and the Department of Health and Human Services Belmont Report. All tumors were frozen after surgical resection. These tumors were either clinically or genetically characterized in the department of neurosurgery of the Pellegrin Hospital (Bordeaux, France), and informed consent was obtained in accordance with the French legislation or was obtained from the processing of biological samples through the Centre de Ressources Biologiques (CRB) Santé of Rennes BB-0033-00056. The research protocol was conducted under French legal guidelines and fulfilled the requirements of the local institutional ethics committee. GBM were classified according to (i) the presence of IDH1, OLIGO2 and TP53 expression and (ii) tumor phenotype (size and form of tumor cells, hyperplasia, necrosis, proliferation index). Primary GBM lines were generated as previously described (Avril et al, 2017). Briefly, fresh tumor tissues were mechanically dissociated using gentle-MACS dissociator following the manufacturer's instructions

(Miltenyi Biotec, Paris, France). RNS cells (neurospheres enriched in cancer stem cells) were directly cultured in DMEM/Ham's F12 (Lonza, Verviers, Belgium) supplemented with B27 and N2 additives (Invitrogen, Cergy Pontoise, France), EGF (20 ng/ml), and FGF2 (20 ng/ml) (Peprotech, Tebu-Bio).

### IRE1 sequencing

IRE1 exon sequencing was performed by Beckman Coulter Genomics (Takeley, UK) using specific primers flanking exonic regions of IRE1. The presence of IRE1alpha mutation was detected using nucleotide sequence alignment software. Tumor in which IRE1 mutation was identified presented classical GBM characteristic with endothelial hyperplasia and MIB1 proliferation index of 15%, and was IDH1 negative, with 5% of OLIG2 and 5% of TP53 positive cells.

### Cloning and site-directed mutagenesis

Selected punctual mutations were introduced on IRE1alpha exonic sequence using QuickChange Directed Mutagenesis kit with the primers described in Appendix Table S6. The wild-type or mutated sequences were then cloned in the multicloning site of the expression lentivector pCDH-CMV-MCS-EF1-Puro-copGFP (System biosciences). The presence of only mutations of interest was checked by a minimum two-X cover sequencing (Beckman Coulter Genomics).

### Cell culture and treatments

U87MG (ATCC) and U251MG (Sigma, St Louis, MO, USA) cells were authenticated as recommended by AACR (http://aacrjournals.org/content/cell-line-authentication-information) and tested for the absence of mycoplasma using MycoAlert® (Lonza, Basel, Switzerland) or MycoFluor (Invitrogen, Carlsbad, CA, USA). U87 cells (ATCC) were grown in DMEM Glutamax (Invitrogen, Carlsbad, CA, USA) supplemented with 10% FBS. U87 were stably transfected at MOI = 0.3 with pCDH-CMV-MCS-EF1-Puro-copGFP (System biosciences) empty vector (EV), pCDH-CMV-MCS-EF1-Puro-copGFP containing IRE1alpha wild-type sequence (WT), or mutated sequence (P336L, A414T, S769F, or Q780*). U87 cells were selected using 2 μg/ml puromycin, and polyclonal populations were tested for GFP expression. Transfections of GBM primary cell lines with IRE1 WT and Q780* were performed using Lipofectamine LTX (Thermo Fisher Scientific), according to the manufacturer's instructions. For microarray experiments, tunicamycin [purchased from Calbiochem (Merck KGaA, Darmstadt, Germany)] was used at 0.5 μg/ml for 16 h. Actinomycin D was purchased from Sigma-Aldrich (St Louis, MO, USA) and used as indicated. For flow cytometry, antibodies against human CD11b, CD31, and CD45 were obtained from BD Biosciences (Le Pont-de-Claix, France). Anti-miR-17 (miRvana) was purchased from Thermo Fisher Scientific.

### Semi-quantitative PCR and Quantitative real-time PCR

Total RNA was prepared using the TRIzol reagent (Invitrogen, Carlsbad, CA, USA). Semi-quantitative analyses were carried out as previously described (Dejeans et al, 2012; Pluquet et al, 2013). PCR products were separated on 4% agarose gels. All RNAs were reverse-transcribed with Maxima Reverse Transcriptase (Thermo

Scientific, Waltham, MA, USA), according to manufacturer protocol. All PCRs were performed with a MJ Mini thermal cycler from Bio-Rad (Hercules, CA, USA) and qPCR with a StepOnePlus™ Real-Time PCR Systems from Applied Biosystems and the SYBR Green PCR Core reagents kit (Bio-Rad). Experiments were performed with at least triplicates for each data point. Each sample was normalized on the basis of its expression of the 18S gene. For quantitative PCR, the primer pairs used are described in Appendix Table S7.

### Western blotting and immunofluorescence analyses

Antibodies are described in Appendix Table S8. All IRE1 signaling analyses were carried out as described previously (Lhomond *et al*, 2015). Cells grown on 22-mm coverslip were washed with PBS, fixed with 4% paraformaldehyde for 10 min at room temperature, and then blocked with 5% BSA, PBS, 0.1% Triton X-100 for 1 h. ER was stained using anti-KDEL antibody (Enzo), and overexpressed IRE1alpha was stained using anti-IRE1alpha antibody (SantaCruz). Cells were incubated with primary antibodies for 1 h at room temperature, washed with PBS, and incubated for 45 min with donkey anti-mouse and donkey anti-rabbit antibodies (Invitrogen). To visualize the nucleus, cells were counterstained with 1 μg/ml 4′,6-diamidino-2-phenylindole (DAPI, Sigma). After mounting, cells were analyzed with a SP5 confocal microscope (Leica Microsystems, Mannheim, Germany).

### Intracranial injections, tumor size, and blood capillary measurements

All animal procedures met the European Community Directive guidelines (Agreement B33-522-2/ No DIR 1322) and were approved by the local ethics committee. Two independent sets of experiments were carried out using 8-week-old male Rag-γ2 mice housed in ventilated racks. The protocol used was as previously described (Auf *et al*, 2010). Cell implantations were at 2 mm lateral to the bregma and 3 mm in depth using seven different sets of cells for U87-EV cells, U87-WT cells, U87-S769F cells, U87-Q780* cells, U87-P336L cells, U87-A414T cells, and U87 IRE1-DN cells. Fifteen days post-injection, or at first clinical signs, mice were sacrificed, brains were frozen, and sliced using a cryostat. Brain sections were stained using H&E staining or anti-vimentin antibodies (Interchim) for visualization of tumor masses. Tumor volume was then estimated by measuring the length ($L$) and width ($W$) of each tumor and was calculated using the following formula ($L \times W^2 \times 0.5$). CD31-positive vessels were numerated after immunohistologic staining of the vascular bed using rat antibodies against CD31 (PharMingen) and fluorescent secondary antibodies (Interchim). Imaging was carried out using a Axioplan 2 epifluorescence microscope (Zeiss) equipped with a digital camera Axiocam (Zeiss). Blood vessels were quantified by two independent investigators using a blinded approach. Vessel number was measured in 12–20 images per condition using ImageJ software. This quantification was made three times for each image, and three vessel sizes (surface) were reported: between 100 pixel² and 500 pixels², more than 500 pixel², or more than 5,000 pixel² (1 pixel = 0.67 μm). The average of vessel number of each size was calculated per brain. Experiments were repeated at least on five Rag-γ2 mice for each condition (up to 15). For GBM primary cell-line implantation, $5 \times 10^5$ cells/

mice were injected in immunocompromised nude mice as described (Avril *et al*, 2017) and above with U87 cells. Mice were daily clinically monitored and sacrificed at the first clinical signs. Mouse brains were collected, fixed in formaldehyde solution 4%, and paraffin-embedded for histological analysis after H&E staining. Tumor burden was compared in the different groups of mice and analyzed using ImageJ software. Furthermore, vascularization of the tumors (CD31), macrophage infiltration (IBA1), and invasion tumor (vimentin) were monitored using immunohistochemistry (Appendix Table S8).

### Statistical analyses

Data are presented as mean ± SD or SEM (as indicated). Statistical significance ($P < 0.05$ or less) was determined using a paired or unpaired *t*-test or ANOVA as appropriate and performed using GraphPad Prism software (GraphPad Software, San Diego, CA, USA).

### Peripheral blood mononuclear cells chemoattraction assay

Peripheral blood mononuclear cells (PBMC) were isolated from healthy donors as described previously (Avril *et al*, 2012). PBMC were washed in DMEM, placed in Boyden chambers ($5 \times 10^5$ cells/chamber in DMEM; Millipore, France) that were placed in DMEM or conditioned medium from cells treated with mirVana miRNA-17 inhibitor (Thermo Fisher Scientific, Waltham, MA, USA), and then incubated at 37°C for 24 h. The migrated PBMC (under the Boyden chambers) were collected, washed in PBS, and cells were stained for monocytes, T, B, and NK cell markers (anti-CD14, anti-CD3, anti-CD19, and anti-CD56, respectively) and analysis by flow cytometry as described below. The relative number of migrated cells was estimated by flow cytometry by counting the number of cells per minute.

### Tumor migration assay

Parental U251 and U87 cell lines transfected with either empty vector pcDNA 3.1/myc-His B or lenti-pCDH-IRE1 A414T, and subsequently transfected with siRNA against XBP1[19] or anti microRNA-17, were washed in DMEM, placed in Boyden chambers ($10^5$ cells/chamber in DMEM) that were placed in DMEM 20% FBS, and incubated at 37°C. After 24 h, Boyden chambers were washed in PBS and cells were fixed in PBS 0.5% paraformaldehyde. Non-migrated cells inside the chambers were removed, and cells were then stained with Giemsa (RAL Diagnostics, Martillac, France). After washes in PBS, pictures of five different fields were taken. Migration was given by the mean of number of migrated cells observed per field. For GBM primary cell migration, Boyden chambers were previously coated with 0.1% collagen solution (Sigma-Aldrich).

### FACS analyses

Glioblastoma multiform specimens were dissociated using the gentleMACS dissociator (Miltenyi Biotec, Paris, France) according to manufacturer's recommendations, and cells were directly used for flow cytometry analysis. Cells were washed in PBS 2% FBS and incubated with saturating concentrations of human

**The paper explained**

**Problem**

Glioblastoma multiform (GBM) is the most lethal primary brain cancer with an overall survival of 15 months and no effective treatment. The endoplasmic reticulum stress sensor IRE1 contributes to GBM progression, impacting tissue invasion and tumor vascularization. IRE1 is an RNase that signals by catalyzing the splicing of the mRNA encoding the transcription factor XBP1 and by regulating the stability of certain miRNAs and mRNAs through a process known as regulated IRE1-dependent decay (RIDD).

**Results**

We found that signaling from IRE1 RNase domain defined specific expression signatures that were relevant in human GBM. This allowed us to demonstrate the antagonistic roles of XBP1 mRNA splicing and RIDD on tumor outcomes.

**Impact**

This study provides the first demonstration of a dual role of IRE1 downstream signaling in cancer and opens a new therapeutic window to impair tumor progression and/or enhance sensitivity to current treatments.

immunoglobulins and fluorescent-labelled primary antibodies for 30 min at 4°C. Cells were then washed with PBS 2% FBS and analyzed by flow cytometry using a FACSCanto II flow cytometer (BD Biosciences). The population of interest was gating according to its FSC/SSC criteria. The dead cell population was excluded using 7-amino-actinomycin D (7AAD) staining (BD Biosciences). Data were analyzed with the FACSDiva (BD Biosciences).

**Data availability**

The data from this publication have been deposited to the Array-Express database (https://www.ebi.ac.uk/; tumor transcriptomes GBMmark) with the accession number E-MTAB-6326 and GEO database (https://www.ncbi.nlm.nih.gov/geo/; cell-line transcriptomes) with the accession number GSE107859.

**Expanded View** for this article is available online.

**Acknowledgements**

We thank Drs S. Manié (Lyon) and R. Pedeux (Rennes) for critical reading of the manuscript, the CRB Santé of Rennes BB-0033-00056 for patient samples, the BIOSIT High Precision histopathology platform H2P2 (A. Fautrel and P. Bellaud; https://biosit.univ-rennes1.fr/?q=en/corefacilities/02/05) for their remarkable work, Biotrial (https://www.biotrial.com/) for help with some animal models and R. Fronzes (IECB, Bordeaux) for help with IRE1 3D models. Affymetrix microarrays were processed in the Microarray Core Facility of the Institute in Regenerative Medicine and Biotherapy, CHU-INSERM-UM Montpellier (http://irmb.chu-montpellier.fr/). This work was funded by grants from Institut National du Cancer (INCa; PLBIO: 2017-148, PLBIO: 2015-111, INCA_7981), La Ligue Contre le Cancer (Comité des Landes, LARGE project), and PHC Maimonide to EC; EU H2020 MSCA ITN-675448 (TRAINERS) and MSCA RISE-734749 (INSPIRED) to EC and AS and from Région Bretagne «AAP CRITT santé 2013» to VQ. SL was funded by a PhD scholarship from the French government and by a scholarship from the Fondation pour la Recherche Médicale. ND was funded by a post-doctoral fellowship from the Fondation de France. M. McMahon was funded by an ARED PhD scholarship from the Région Bretagne. This work was carried out under the auspices of REACT (REnnes brAin Cancer Team).

**Author contributions**

SL, ND, NP-L, KB, OPl, and EC carried out experiments on U87 or U251 cells and studied the mouse *in vivo* data. TA, DD, GJ, and JO contributed the work on primary GBM lines. MMc, KS, MMa, HB, and AS contributed the work on microRNA. KV, OPa, ML, and AC performed all the bioinformatics analyses. RP and FJ were in charge of the animal experiments. P-JLR, EV, VQ, AE, and JM contributed the GBMmark cohort. JBP provided the IRE1 RNase inhibitor. CH contributed unpublished information and reagents and helped mature the manuscript. SL, ND, OPl, TA, and JO interpreted the data. EC conceived and realized the study. AC, AS, CH, TA, and EC wrote the manuscript.

**Conflict of interest**

The authors declare that they have no conflict of interest.

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
