## [Review Process File · EMBO Molecular Medicine]

Dual IRE1 RNase functions dictate glioblastoma development

Stéphanie Lhomond, Tony Avril, Nicolas Dejeans, Konstantinos Voutetakis, Dimitrios Doultzinos, Mari McMahon, Raphaël Pineau, Joanna Obacz, Olga Papadodima, Florence Jouan, Heloise Bourien, Marianthi Logotheti, Gwénaële Jégou, Néstor Pallares-Lupon, Kathleen Schmit, Pierre-Jean Le Reste, Amadine Etcheverry, Jean Mosser, Kim Barroso, Elodie Vauléon, Marion Maurel, Afshin Samali, John B. Patterson, Olivier Pluquet, Claudio Hetz, Véronique Quillien, Arisotelis Chatziioannou and Eric Chevet

Review timeline:

Submission date:	24 April 2017
Editorial Decision:	24 May 2017
Additional correspondence (author)	15 June 2017
Revision received:	10 October 2017
Editorial Decision:	05 December 2017
Revision received:	08 December 2017
Accepted:	11 December 2017

Editors Robert Buccione and Céline Carret

Transaction Report:

1st Editorial Decision

24 May 2017

Thank you for the submission of your manuscript to EMBO Molecular Medicine.

I apologise for the delay in providing you with a decision. On one hand we experienced difficulties in securing willing and appropriate reviewers and then obtaining their evaluations in a timely fashion. In addition, reviewer 2 withdrew from the process and I therefore had to resort to an additional reviewer, which further delayed the process.

As you will see, the reviewers display very different degrees of enthusiasm. While reviewer 1 is globally quite positive, reviewer 3 is rather unenthusiastic if a bit cursory, and reviewer 4 sits in a middle ground.

Overall, a major criticism that emerges is that the study is based on a single, established cell line. I definitely agree that this is indeed an important concern that compromises the translational relevance of the manuscript (which is quite important for our title). Another important issue is the unclear case for relevance of IRE1 mutations in GBM, especially given that at least in one of the biggest collection of GBM exome data (TCGA) no IRE1 mutations have been observed. Furthermore, the fact that the analysis is based on microarray data, when better resolution on splicing of different transcripts could have been achieved by mining RNAseq data, was also mentioned as a concern.

It is also clear that there is a lack of information on experimental procedures, the materials and animals used, and in general the need to take advantage of, and analyse previously available datasets. There is also a clear requirement to better discuss crucial aspects of the work including its translational implications.

During our reviewer cross-commenting exercise, reviewer 1 came to agree on the need to address the concern on the single cell line and all agreed that this should be addressed preferably as reviewer 4 suggests, with patient-derived cell lines. I should add that there was also a clear concern raised on the feasibility of such an undertaking within an acceptable timeframe.

In conclusion, given the potential interest of your work, I am prepared to consider a substantially revised submission, with the understanding that the reviewers' concerns must be fully addressed. Clearly, the manuscript could be potentially very much improved by extensive revision and experimentation, but with no guarantee that the currently unsupported main conclusions would remain valid. For this reason, and to save you from any frustrations in the end I would strongly advise against returning an incomplete revision and would also understand your decision if you chose to rather seek rapid publication elsewhere at this stage. I would appreciate a note informing me of your decision on how you wish to proceed in this respect.

***** Reviewer's comments *****

Referee #1 (Remarks):

Lohmond and colleagues have assembled an interesting and complex manuscript providing additional insight into the roles of IRE1, an endoplasmic reticulum(ER)-localized kinase/RNase, in the development of glioblastoma multiform (GBM). The RNase activity of IRE1, a central player in the ER stress-activated unfolded protein response (UPR), is essential for the expression of XBP1s, a transcription factor. Some XBP1s target genes promote cancer cell survival and tumor progression. In addition, the RNase activity of IRE1 mediates degradation of various mRNA and miRNA, a process termed regulated IRE1 dependent decay (RIDD). Previous work has implicated RIDD in tissue invasion, growth and vascularization of GBM. In the current study, a combination of GBM samples and cell lines engineered to express GBM-related IRE mutants were used to generate evidence indicating that XBP1s promotes tumor progression while RIDD slows tumor development. In this way, distinct IRE1-mediated signaling outcomes appear to differentially influence GBM cell biology, GBM aggressiveness, and the prognosis for GBM patients. In total, the work is well-performed and the data support the authors' conclusions. Specific comments and questions are as follows:

1. The data concerning the various IRE1 mutants is not entirely straightforward. In particular, it seems difficult to interpret the analysis of cells expressing the P336L mutant, especially due to its extremely high levels of p53. Given the volume and complexity of the data in the manuscript, is all of it essential to the story?
2. The manuscript would be improved by inclusion of a better description of the methods used to classify tumors and cell lines as high or low for IRE1 activity, XBP1s activity, RIDD activity, etc. While such details may be provided in other publications, a bit more information in the current manuscript seems appropriate given that these various classifications are critically important in these studies.
3. The authors use the word "antagonistic" to describe the effects of distinct IRE1 RNase functions - XBP1 mRNA splicing and RIDD - in GBM development. However, to this reviewer, it seems that these two IRE1 RNase functions simply exert different overall influences on GBM cells and tumors. XBP1s and RIDD do not appear to interfere with each other.

Referee #3 (Comments on Novelty/Model System):

All the analyses were done in a single cell line.

Referee #3 (Remarks):

Llomond et. al conducted a study on the RNase function of IRE1 and impact on GBM. The topic is highly important due to the fact that IRE1 is often mutated in GBM and the result of these mutations are not clearly understood. However, the article fails to deliver a convincing story and is very hard to follow.

The main problem is the fact that the entire study was conducted in a single cell line. The use of constructs over expressing mutated IRE1 instead of CRISPR derived mutants is another problem.

I am puzzled by the result of the xenografts and cannot make a logical connection to the scenario found in GBM patients.

It is very unlikely that IRE1 affects the splicing of a single gene. Authors should have checked their RNAseq data more carefully for other splicing alterations.

A more comprehensive gene expression correlation analysis in GBM for IRE1 should have been done.

How the different mutations in IRE1 affect survival?

The authors should have shown mir-17 and IRE1 correlate in tumors.

Discussion is mostly a summary of results. The impact of their finding to therapy is not discussed properly.

Minor

Figure 2B. It is necessary to show a diagram displaying the differences between the u and s isoforms of XBP1.

Figure 3B is impossible to read.

Referee #4 (Comments on Novelty/Model System):

Authors have chosen to use just one long-term established cell line for their entire experimental work. The addition of a patient-derived cell line is necessary to validate their claim.

Referee #4 (Remarks):

Overall, this is very exciting work by Lhomond et.al. In this manuscript authors, for the first time, demonstrated the antagonistic function of IRE1 RNase in GBM tumor development and progression. Authors also identified a novel "driver" mutation in IRE1 and validated the role of this mutation using some functional assays. On one hand, manuscript describes very novel functional consequence of IRE1 RNase function; there are some major drawbacks in the manuscript that needs to be addressed before it is accepted for publication. Additionally, the way manuscript is written is very confusing, it is not clear if authors want to go after functionality of mutated IRE1 or expression of IRE1. Some of the major issues in the manuscript are listed below:

1. In most current data from TCGA cohort (www.cbiportal.org) which comprises of 1102 samples including lower grade glioma, only one mutation in IRE1 was observed, which is, K633Q. Even more surprisingly, this mutation was observed only in one lower grade glioma sample. None of 276 GBM samples show the mutation in IRE1. Authors are requested to elaborate their thoughts on such a discrepancy. Authors are also requested to elaborate their reliance on old genomics data (2007 - 2008) when more advanced NGS data is freely available for analysis of mutations?
2. Authors are also requested to explain in detail, how low and high expressions for IRE1, XBP1, RIDD, etc. were determined? What dataset were used? Did they use microarray data? If Yes, why? And why not use RNAseq data, which gives much better resolution for transcriptomic analysis?
3. For intracranial studies, it is not clear from methods, what mouse strain was used. It is very critical to understand the mouse strain utilized for xenograft study to validate authors claims of macrophage infiltration.
4. Authors are requested to explain Figure S6B in detail - it was not clear from the data that why would decrease in the area suggest neurosphere formation?
5. Authors are requested to get IHC stains of IBA1 and XBP1 scored by a board-certified pathologist. It will add much more conviction to authors claim.
6. For such a novel claim, authors are requested to include the data from patient-derived cell lines (at least one more) rather than just one long-term established lines

Additional correspondence (author)

15 June 2017

Subject: Experimental plan to address reviewers' issues – EMM2017

Thank you for your response and for efficiently handling our manuscript. Please find below our experimental plan and the anticipated time-line to address the reviewers' comments (experimentally). In addition to those we will of course discuss several points that either do not need or cannot be addressed experimentally for the current manuscript.

Again, we think that our manuscript provides new functional information about the role of IRE1 signaling in human GBM and that the data we provide point towards the identification of IRE1 as a genuine therapeutic target in GBM and more generally in cancers.

Our results are also in accordance with those obtained by our collaborators (Samali Lab) in triple negative breast cancers, and indicate that the most aggressive tumors in both cancers exhibit high XBP1s activity linked to the mesenchymal and pro-inflammatory tumor phenotypes.

The combination of our initial set of data and the ones we propose to add herein will not only delineate a novel classification of GBMs based on IRE1 activity status but also provide the rationale for targeted therapeutic approaches using IRE1 RNase inhibitors.

One major comment from the reviewers notes that in our expression system there is residual signaling from endogenous IRE1 and that a single cell line was used as our model for the in vitro experiment. To address these concerns, we propose to add a significant amount of complementary data obtained from primary GBM cell lines, provided in the attached figures. Importantly, all mutations found in cancer cases were heterozygous, indicating that their adverse effects on glioma progression occur in the presence of one copy of wild-type IRE1.

1) Classification of human primary GBM lines

1.1. Transcriptome analyses and scoring according to the initial approach carried out in human tumors.

In the lab, we have generated several primary cell lines from GBM specimens growing as adherent cells or neurospheres in culture. Using these primary human GBM lines, human primary GBM tumors, human neural progenitors and astrocytes we first generated transcriptomic data and then used scoring system described in the original manuscript.

Figure 1: Molecular classification of human primary GBM lines - A) Classification of tumors and their corresponding primary cell lines, grown in neurospheres, and in adherence conditions. Tumors and primary cell lines were scored for their XBP1 and RIDD activities using transcriptome data (as described in our manuscript). RNS, .NS and .BN for neurospheres cell lines; .SVF for adherent cell lines; .tum for tumor specimen; astro for primary astrocytes; prog for neural progenitor cells. B) Classification of the lines grown as neurospheres based on their transcriptome. Scoring of the XBP1 and RIDD pathways was carried out as reported in our original manuscript and the 12 lines were classified as belonging to the 4 groups defined as a) XBP1+/RIDD-, b) XBP1+/RIDD+, c) XBP1-/RIDD+ and d) XBP1-/RIDD-.

This approach allowed us to classify these samples into 4 categories based on their IRE1 signaling profile: XBP1+/RIDD+; XBP1+/RIDD-; XBP1-/RIDD+; XBP1-/RIDD- (**Figure 1A**). These data indicated that tumor lines and tumor specimens exhibit very heterogeneous patterns regarding IRE1 activation but also that tumors differ from normal astrocytes (which do not appear to exhibit strong IRE1 signaling properties). Of note is the fact that neural progenitors clustered in the XBP1-/RIDD+ group suggesting that RIDD but not XBP1 could play a role in cell differentiation.

Using the same approach we then categorized the 12 primary lines (enriched in cancer stem cells; RNS lines) grown as neurospheres. These 12 lines also clustered into the four categories described above. Four of these lines fell into the XBP1+/RIDD- group, 2 into the XBP1+/RIDD+ group, 3 within the XBP1-/RIDD+ group and 3 into the XBP1-/RIDD+ group (**Figure 1B**). As a

consequence the 12 RNS lines provide us with a powerful tool to investigate the role of IRE1 in primary GBM lines.

This work has been finalized and a formalization of the data is currently under way in collaboration with the bioinformatics team (A. Chatziioannou).

1.2. Phenotypic characterization in terms of UPR (markers, IRE1 signaling, RIDD, XBP1s, miR17), IRE1 sequencing ... presence of mutations?

Persons involved: Tony Avril (scientist); Mari McMahon (PhD Student); Héloïse Bourien (MSc student); Gwénaële Jégou (Tech).

We are now further documenting the phenotype of those tumor cells by analyzing the expression levels of IRE1 and the mutational state of the ERN1 gene in each line. Moreover, we will evaluate the same markers that were analyzed in U87 cells and their IRE1 mutant expressing counterparts (IRE1 phosphorylation, XBP1 mRNA splicing, expression levels of RIDD targets, miR-17 and targets expression). This analysis will allow us to qualitatively and quantitatively compare the IRE1 signaling status in the primary lines to that observed in U87 cells and U87 expressing IRE1 variants.

Figure 2: Phenotype of primary GBM lines. A) Classification of the lines grown as neurospheres based on their transcriptome as in Figure 1. B) Phase contrast images of 4 lines in culture. C) Sensitivity of the same four lines to the standard chemotherapy for GBM, temozolomide.

To also evaluate the relevance of these molecular signatures to the phenotypes and to the sensitivity of the primary lines to treatment with temozolomide, we investigated the phenotype of select lines in the XBP1+/RIDD-, XBP1-/RIDD-, and the XBP1+/RIDD+ group. Interestingly, the lines that belonged to the XBP1+/RIDD- group (85, 130) exhibited an adherent phenotype that may correlate with the mesenchymal features seen in the tumors exhibiting high IRE1 activity (as seen in the original version of our manuscript) (*Figure 2A,B*). Moreover, when XBP1s activity was low (96), or counterbalanced by a strong RIDD activity (87), these mesenchymal features were attenuated (*Figure 2B*). Finally, in contrast to the other lines, the lines belonging to the XBP1+/RIDD- group were found to be highly resistant to a treatment with the alkylating agent temozolomide (TMZ), which also known to induce ER stress (*Figure 2C*).

This work will take another 2 months to be finalized

1.3. Phenotypic characterization in terms of secretome and capacity to chemoattract PBMCs, induce angiogenesis, and ability to migrate.

Persons involved: Tony Avril (scientist); Joanna Obacz (post-doc); Gwénaële Jégou (Tech).

Our goal in this section is to provide further information on the phenotypes exhibited by primary lines. First of all, we have already observed a significant difference in migration capacities between primary cell lines from XBP1+/RIDD- and XBP1-/RIDD+ groups (see preliminary data in *Figure 3*). Correlations between the IRE1 classification of GBM primary cell lines and differential

expression of chemokines/cytokines involved in immune cell chemoattraction will be next explored using our transcriptome data.

Figure 3: Example of the phenotypic characterization of the primary lines. A) Migration properties of the lines as explored using Boyden chambers. The lines tested were classified according to the profiles presented in Fig 3B – top panel. B) Expression of VEGFA mRNA in 4 lines distributing in the four groups.

We have investigated the mRNA expression levels of chemokines in primary lines as illustrated for VEGFA and corroborated these data with the IRE1 signaling group. We now have to i) validate these data at the protein level and ii) perform the functional assays.

This work will take another 3-4 months to be finalized

1.4. In vivo --- orthotopic grafts evaluate tumor size and mouse survival. Characterize the nature of tumor stroma (infiltrate, angiogenesis) and reconcile with IRE1 signaling characteristics.
 Persons involved: Raphael Pineau (engineer); Véronique Quillien (senior investigator)

When these primary lines were injected orthotopically in mouse brain, the tumors that formed were very large if they were in the XBP1+/RIDD- classification, or morphologically smaller in the three other groups (**Figure 4A**).

Figure 4B describes the quantification of tumor size, and indicates that tumors derived from lines belonging to the XBP1+/RIDD- group were indeed the largest and the most aggressive (**Figure 4C**). Altogether, these data provide a framework for an in vitro model that could recapitulate some of the characteristics of the primary tumors identified in our manuscript and their status of IRE1 pathway activation.

Figure 4: In vivo characterization of the primary lines following injection in the brain of immunocompromised mice. A-C) Lines from the a, b and d groups were injected into the brains of immunocompromised mice. Tumors were analyzed by vimentin staining at sacrifice, 4 examples are provided (A), tumor size was quantified (B) and mouse survival determined (C).

We now aim at analyzing the tumors for their angiogenic status, their infiltration status mainly by macrophages as the immunocompromised mouse model was used for our in vivo experiments.

Immunohistochemistry reactions have been performed and we are now analyzing the data. This work should take another 1.5 month to be completed.

2) Sequence analyses

Persons involved: Aristotelis Chatziioannou (PI); Tony Avril (scientist); Olga Papadodima (scientist); Konstantinos Voutetakis (PhD student).

1) formalizing the scoring method to classify tumors (for IRE1 +/- but also for the 4 groups which include a scoring step). Herein we propose to formalize the scoring methodology that was used in the initial version of the manuscript.

2) use this to test new RNAseq datasets obtained on GBM (ICGC).

3) In RNAseq data, correlate the results obtained with our signature approach to the analysis of XBP1 splicing and consolidate the approach.

4) As IRE1 mutations are not that frequent, new genome/exome sequence data could be reanalyzed to expand the spectrum (at least at informational levels)

5) address the issue of other IRE1-dependent splicing targets. Although this point will go against 25 years of work (including published articles from the Walter group demonstrating the unicity of this event), we need to somehow evaluate this for instance by evaluating the presence of novel IRE1 cleavage sequences. Although this is of interest and might be discussed in the rebuttal, it really stands outside the scope of the study

This work will take another 5 months to be finalized

3) Alteration of IRE1 signaling in primary GBM lines

Persons involved: Dimitrios Doultinos (PhD student); Joanna Obacz (post-doc); Gwénaële Jégou (Tech); Héloïse Bourien (MSc student).

In order to test the impact of IRE1 activity modulation on primary GBM lines, we propose to genetically alter IRE1 in the selected cell lines and evaluate the resulting phenotypes using recombinant expression of IRE1 wild type, Dominant Negative (DN) as well as the mutants described in our study.

Figure 4: Expression of wt and mutant IRE1 in primary GBM lines.

Stably modified primary lines will then be investigated for their UPR signaling properties as carried out on the primary lines (see section 1.2) and their ability to exhibit select phenotypes in vitro such as migration, cytokine production (to induce migration of PBMCs or angiogenesis) as performed on the parental lines (see section 1.3). Finally the characteristics of the IRE1-modulated primary lines to resist to ER stress (Tunicamycin) or TMZ treatment will also be investigated.

This work will take another 5 months to be finalized

In summary, with the addition of the primary cell data (that will take around 5 months to finalize), we strongly believe that our manuscript will provide the information that was noted by the reviewers as missing in the original version. This will convincingly prove the identification of IRE1 as a key player in GBM development and aggressiveness and demonstrate its value as a therapeutic target in this dismal disease.

1st Revision - authors' response

10 October 2017

Referee #1 (Remarks):

We thank reviewer 1 for the constructive comments on our manuscript.

Referee 1 comment 1. The data concerning the various IRE1 mutants is not entirely straightforward. In particular, it seems difficult to interpret the analysis of cells expressing the P336L mutant, especially due to its extremely high levels of p53. Given the volume and complexity of the data in the manuscript, is all of it essential to the story?

We thank the reviewer for this remark. We agree that the amount of data provided in the manuscript is very dense. We have tried our best in the revised manuscript to reduce this information burden and focus on the take-home message. Most of the figures were rebuilt to make them more intelligible and the text was also revised for the sake of simplicity.

Concerning the link of P336L with p53, we believe that this observation explains why U87 cells expressing this IRE1 variant do not form any tumor in vivo. The data relative to p53 were now placed in the supplemental information section in the revised version of the manuscript (**new Figure S5**).

Referee 1 comment 2. The manuscript would be improved by inclusion of a better description of the methods used to classify tumors and cell lines as high or low for IRE1 activity, XBP1s activity, RIDD activity, etc. While such details may be provided in other publications, a bit more information in the current manuscript seems appropriate given that these various classifications are critically important in these studies.

A clearer description of our methodology has now been added in the revised supplemental Materials and methods of our manuscript on p2-5, especially the details of the statistical tests applied to validate the methodologies used are provided. Moreover, we have added one supplemental figure (**new Figure S1**) that depicts our approach and provides the list of the 38 IRE1-hub genes used as an IRE1 activity signature throughout the manuscript. Finally, another piece of evidence/validation of our approach is provided by the use of the TCGA RNAseq dataset. Indeed, we correlated the enrichment of XBP1 mRNA splicing events in the tumors scored to belong to the XBP1shigh (XBP1+) groups (**new Figure S7D**), thereby increasing the strength of our classification.

Referee 1 comment 3. The authors use the word "antagonistic" to describe the effects of distinct IRE1 RNase functions - XBP1 mRNA splicing and RIDD - in GBM development. However, to this reviewer, it seems that these two IRE1 RNase functions simply exert different overall influences on GBM cells and tumors. XBP1s and RIDD do not appear to interfere with each other.

We thank the reviewer for this comment. Our data indicate that XBP1s has a positive impact on macrophage recruitment to the tumor, as well as on angiogenesis and cell invasion/migration. In contrast, we show that RIDD activity might antagonize the effects of XBP1s selectively on angiogenesis and on cell migration/invasion. We agree that the word "antagonistic" used in the title might be too strong regarding the indirect nature of IRE1 signaling on the biological effects quantified. This is why we replaced "antagonistic" by "dual" in the title of the revised manuscript.

**Referee #3 (Comments on Novelty/Model System):
All the analyses were done in a single cell line.**

We thank the reviewer for this comment. To address this point, we have now added two more figures that validate our model in primary GBM lines (shown in the **new Figures 7 and 8**). First, the signatures for IRE1 signaling (XBP1s and RIDD) were confronted to the transcriptome data obtained from 12 primary GBM lines and revealed that the lines actually clustered in four groups, namely XBP1s+/RIDD+; XBP1s+/RIDD-; XBP1s-/RIDD-; XBP1s-/RIDD+, and elicited specific IRE1 signaling properties (see Figure R1 of the rebuttal). These data are now shown in the **new Figure 7** of the revised manuscript. Moreover, when the primary lines were tested for their tumorigenic potential in vivo, this revealed that as observed in patients' tumors, high XBP1s (XBP1+) correlated with high aggressiveness (**new Figure 7E**). Finally, another correlation was made for the capacity of tumors to recruit monocytes/macrophages. Indeed, a good correlation between XBP1s and IBA1 (**new Figure 4D-E**) staining in human tumors was observed and tumors initiated by cells belonging to the XBP1+ groups were highly infiltrated by mouse macrophages (**new Figure 7H**).

Figure R1: Molecular classification of human primary GBM lines - Classification of tumors and their corresponding primary cell lines, grown as neurospheres or as adherent cells. Tumors and primary cell lines were scored for their XBP1 and RIDD activities using transcriptome data (as described in our manuscript). RNS for neurospheres cell lines; RADH for adherent cell lines; TUM for tumor specimen; ASTRO for primary astrocytes; PROG for neural progenitor cells. For the scoring of XBP1s were used only 17/40 genes and for the scoring of RIDD 24/38; the remaining genes were not found after the annotation of Agilent Probe IDs.

Second, to demonstrate that modulating IRE1 activity in those lines could actually impact on their phenotypes, we either overexpressed a wild type (WT) form of IRE1 or a RNase defective variant (Q780*) that impairs XBP1 mRNA splicing (shown in the **new Figure 2** in U87 cells). We show in the **new Figure 8** that overexpression of WT IRE1, which happens to increase IRE1 activity, also increases tumor cell migration and chemokine production (at both mRNA (**new Figure 8E**) and protein levels, as seen by the capacity of conditioned media to attract monocytes, (**new Figures 8H, I**)). Impairing IRE1 activity through the overexpression of the Q780* variant also impaired IRE1-dependent cell migration and monocyte chemoattraction (**new Figure 8**).

Collectively, the new data presented in the revised manuscript reconcile the phenotypes observed in human tumors and in vitro/vivo experiments carried out in the laboratory. From our experiments, we demonstrate that XBP1+ tumors exhibit a very aggressive phenotype and that RIDD might tune it down.

Referee #3 (Remarks):

Llomond et. al conducted a study on the RNase function of IRE1 and impact on GBM. The topic is highly important due to the fact that IRE1 is often mutated in GBM and the result of these mutations are not clearly understood. However, the article fails to deliver a convincing story and is very hard to follow. The main problem is the fact that the entire study was conducted in a single cell line. The use of constructs over expressing mutated IRE1 instead of CRISPR derived mutants is another problem.

We have worked on the manuscript to make the story easier to follow. The figures were extensively reshaped (all the figures were modified except for the new Figure 3) and the text was also modified to clarify the objectives and make the logical flow more robust throughout the manuscript. In addition, by adding more details on the procedures used to analyze tumor transcriptomes, we believe that the manuscript is easier to understand and more precise. Concerning the issue regarding the use of a single cell line, we now present in the revised manuscript two additional figures that report IRE1 (XBP1s and RIDD) signatures in 12 primary GBM lines (see above) and we correlate this information with phenotypes in vitro (migration) and in vivo (tumor formation, angiogenesis, macrophage recruitment). These data are presented in the **new Figure 7** of the revised manuscript. Moreover, we have genetically modified four of these lines to alter IRE1 signaling and found that enhancing IRE1 activity prompted cell migration and the release of chemoattractants whereas blocking IRE1 activity had an opposite effect (**new Figure 8**). We attempted to generate IRE1 mutant and XBP1 deficient primary lines using the CRISPR/CAS9 technology. All our efforts were unsuccessful until now, and considering the doubling time of those

lines, it was not possible to provide this information within the time frame allotted for the rebuttal. This is the reason why we provide the information with the overexpression system. Another point also concerns the fact that the IRE1 variants were all identified as heterozygous, thereby co-existing with the WT allele. As such, the overexpression system would still provide some relevant information compatible with the data observed in human tumors.

Referee 3 comment 1. I am puzzled by the result of the xenografts and cannot make a logical connection to the scenario found in GBM patients.

We agree with this reviewer that the data obtained with U87 do not properly match the results obtained with human tumors. However, these data demonstrate that the expression of IRE1 variants impacts on the phenotypes of U87-derived tumors, by altering their aggressiveness (mouse survival) and their capacity to reshape the tumor stroma (angiogenesis and macrophage infiltration). The data generated with U87 also demonstrate the limit of this model. To avoid such biased approach, we have used primary GBM lines derived from patients' tumors. Using those primary lines, we now demonstrate the relationships between IRE1 signaling (XBP1s and RIDD) and tumor phenotype (including aggressiveness and infiltration by macrophages), results that are consistent with the data obtained in human tumors.

Referee 3 comment 2. It is very unlikely that IRE1 affects the splicing of a single gene. Authors should have checked their RNAseq data more carefully for other splicing alterations.

IRE1 is an ER resident RNase that is conserved from yeast to mammals. It has been identified 20 years ago by the groups of Peter Walter and Joseph Sambrook (with Kazutoshi Mori). The role of IRE1 was first identified to contribute to the non-conventional (spliceosome independent) splicing of HAC1 mRNA in *S. cerevisiae*, a gene that has no homolog in mammals. It was demonstrated at that time that HAC1 splicing was the only RNA splicing event driven by IRE1 in *S. cerevisiae* by the group of Peter Walter. In 2001 the functional homolog of HAC1 was identified as XBP1 in metazoans simultaneously by the groups of Kazutoshi Mori, Randal Kaufman and David Ron. This occurred through a different mechanism that in yeast, involving the removal of a non-conventional intron of 26 nucleotides, thereby leading to a change in the mRNA open reading frame. Until 2006, XBP1 mRNA splicing was the only event believed to be driven by IRE1 in metazoans until the group of Jonathan Weissman (with Julie Hollien) demonstrated that IRE1 could also contribute to mRNA degradation, an activity named Regulated IRE1 Dependent Decay of mRNA. More recently, the tRNA ligase RtcB was identified as the possible ligase whose function combined with IRE1 RNase activity is responsible for XBP1 mRNA non-conventional splicing. Thus far, there is no evidence that other non-conventional splicing events can take place in mammalian cells upon IRE1 activation. In *S. cerevisiae*, it has convincingly been demonstrated by Peter Walter's group that HAC1 mRNA splicing was unique. It is true that we could reanalyze RNAseq data in view of our tumor scoring approach, for unexpected/novel splicing events but this stands beyond the scope of the current manuscript.

Referee 3 comment 3. A more comprehensive gene expression correlation analysis in GBM for IRE1 should have been done.

This is now presented in the revised Figures 1, 4, 6 and S2, S6. The approach used here has allowed us to analyze the entire population of patients instead of using only those exhibiting highest variance. As such the statistical power of our analysis has been improved.

Referee 3 comment 4. How the different mutations in IRE1 affect survival?

Four GBM patients were found to bear mutations on the IRE1 gene. Among these patients, the survival information was only available for one. As a consequence, we cannot evaluate the direct impact of IRE1 mutations on patient survival. We have used the U87 model (presented in the new Figure 3) to assess the impact of mutations on survival and found that the A414T mutations conferred U87 cells with high aggressiveness and yielded lower survival in mice.

Referee 3 comment 5. The authors should have shown mir-17 and IRE1 correlate in tumors.

This information is shown in the revised version of the manuscript in the new Figure 5. We correlate the expression of miR-17 and the activity of IRE1 in human tumors and linked to patients' survival. This information is described on p10-11 of the revised manuscript.

Referee 3 comment 6. Discussion is mostly a summary of results. The impact of their finding to therapy is not discussed properly.

The impact of our finding to potential therapeutic approach is now discussed in the discussion section of the revised manuscript on p14-15.

Minor

Referee 3 comment 7. Figure 2B. It is necessary to show a diagram displaying the differences between the u and s isoforms of XBP1.

Figure R2: Schematic representation of XBP1 splicing. Consequences at the mRNA and protein levels. On the left-hand side of the corresponding PCR amplicons are shown with a difference of 26 nucleotides upon treatment with the ER stressor tunicamycin. (Figure adapted from Dejeans et al. 2012 and from Nagashima et al. 2011).

We provide this reviewer with a schematic representation of XBP1 mRNA splicing and the corresponding results visualized using RT-PCR. The removal of the non-conventional intron of 26 nucleotides by the combined action of IRE1 (RNase) and RtcB (Ligase) leads to the formation of XBP1s mRNA. This information has been extensively documented over the past 15 years in many research articles and reviews. As our manuscript is already very dense we believe that this type of knowledge can be found elsewhere.

Referee 3 comment 8. Figure 3B is impossible to read.

This has been fixed in the revised manuscript.

Referee #4 (Comments on Novelty/Model System):

Authors have chosen to use just one long-term established cell line for their entire experimental work. The addition of a patient-derived cell line is necessary to validate their claim.

This has now been added in the manuscript in the **new Figures 7 and 8**. See also response to **“Comments on novelty/model system” from referee 3**.

Referee #4 (Remarks):

Overall, this is very exciting work by Lhomond et.al. In this manuscript authors, for the first time, demonstrated the antagonistic function of IRE1 RNase in GBM tumor development and progression. Authors also identified a novel "driver" mutation in IRE1 and validated the role of this mutation using some functional assays. On one hand, manuscript describes very novel functional consequence of IRE1 RNase function; there are some major drawbacks in the manuscript that needs to be addressed before it is accepted for publication. Additionally, the way manuscript is written is very confusing, it is not clear if authors want to go after functionality of mutated IRE1 or expression of IRE1. Some of the major issues in the manuscript are listed below:

We thank this reviewer for the nice comments, we have done our best to reformulate our objectives in the manuscript and make the story easier to follow. In the revised version of the manuscript, we have added two new figures and completely revised 5 figures out of the remaining 6. The text has also been worked out in order to clarify our point and make the story more convincing.

Referee 4 comment 1. In most current data from TCGA cohort (www.cbioportal.org) which comprises of 1102 samples including lower grade glioma, only one mutation in IRE1 was observed, which is, K633Q. Even more surprisingly, this mutation was observed only in one lower grade glioma sample. None of 276 GBM samples show the mutation in IRE1. Authors are requested to elaborate their thoughts on such a discrepancy. Authors are also requested to

elaborate their reliance on old genomics data (2007 - 2008) when more advanced NGS data is freely available for analysis of mutations?

This is a very good remark. We have indeed used “old” data sets and did not introduce the new mutations in the course of our project since the experimental pipelines were actually too heavy to do so. As this reviewer will see in the new version of our manuscript, we have only used the properties of the IRE1 mutants identified previously to alter (artificially) IRE1 activity and to yield transcriptional and posttranscriptional cell reprogramming. This has allowed us to come up with XBP1 and RIDD gene expression signatures that were in turn used to deconvolute IRE1 signaling in tumors. We find that IRE1 activity is, as expected, altered in human tumors, most likely due to the tumor intrinsic properties (including external (environment) or internal (oncogene, metabolism) stress) and that these properties are conserved in primary tumor lines. As a consequence, we remain convinced that IRE1 mutations do exist however, these tumors are rare and do impact on a minimal number of tumors. Our findings indicate that IRE1 activity i) is definitely altered in tumors, ii) impacts on tumor properties (correlated in human tumors, and tested in primary GBM lines) and consequently iii) could be used as therapeutic targets in certain classes of GBM. Moreover, regarding the presence of cancer driver mutations in IRE1, our data show that the P336L mutation, which is the only mutation found in more than one tumor (also found in intestinal tumors) can only provide a selective advantage to the tumor cells on a p53 mutant background (which is not the case of U87 cells), thus leading us to conclude that it is not a driver mutation.

Referee 4 comment 2. Authors are also requested to explain in detail, how low and high expressions for IRE1, XBP1, RIDD, etc. were determined? What dataset were used? Did they use microarray data? If Yes, why? And why not use RNAseq data, which gives much better resolution for transcriptomic analysis?

This is now fully detailed in the manuscript. We have, as requested by this reviewer, added the RNAseq data set and all the new data are now presented in the **new Figure 6** of the revised manuscript. These data are consistent with those obtained with the TCGA microarray data and with the our local (GBMmark) microarray data. Using those datasets, we demonstrate that XBP1+/RIDD-tumors are much more aggressive than XBP1-/RIDD+ tumors (presented in the **new Figure 6**). Moreover, using the RNAseq dataset, we show that tumors scored to be in the XBP1+ group with our method exhibit, as expected, a higher number of reads corresponding to the spliced form of XBP1 (see **new Figure S7D**). Collectively, we thank this reviewer for suggesting to add the RNAseq data to our manuscript as this confirmed our results obtained with microarray approaches and also allowed us to confirm the robustness of our approach in identifying XBP1+ tumors.

Referee 4 comment 3. For intracranial studies, it is not clear from methods, what mouse strain was used. It is very critical to understand the mouse strain utilized for xenograft study to validate authors claims of macrophage infiltration.

This has now been added in the methods section on p19 of the revised manuscript.

Referee 4 comment 4. Authors are requested to explain Figure S6B in detail - it was not clear from the data that why would decrease in the area suggest neurosphere formation?

As shown previously by our group (for instance see Dejeans et al. *J Cell Sci.* 2012 Sep 15;125(Pt 18):4278-87 and/or Avril et al. *Clin Cancer Res.* 2017 Sep 22. pii: clincanres.1549.2017), decrease in sphere area coincides with an increased neurosphere formation (provided that the number of cells seeded is identical and that all the cells are aggregated to the sphere) when sphere density is increased. In the time-frame of the experiment, cell survival is allowed only if cells adhere to each other and consequently the density of the spheres somehow reflects the strength of cell-cell interactions within this sphere. As such, this is why we reached the conclusions presented in the manuscript.

Referee 4 comment 5. Authors are requested to get IHC stains of IBA1 and XBP1 scored by a board-certified pathologist. It will add much more conviction to authors claim.

All our immunohistochemically stained slides were read in a blind manner by our collaborator Dr D Chiforeanu from the dept of pathology of the University hospital of Rennes. All his observations converged towards the same conclusions that those reached using image analysis.

Referee 4 comment 6. For such a novel claim, authors are requested to include the data from patient-derived cell lines (at least one more) rather than just one long-term established lines.

To address this point, we have now added two more figures that validate our model in primary GBM lines. First, the signatures for IRE1 signaling (XBP1s and RIDD) were confronted to the transcriptome data obtained from 12 primary GBM lines and revealed that the lines actually clustered in four groups, namely XBP1s+/RIDD+; XBP1s+/RIDD-; XBP1s-/RIDD-; XBP1s-/RIDD+, as did the parental tumors. This is now shown in the **new Figure 7** of the revised manuscript (**see also response to Referee 3**). To further test whether the modulation of IRE1 activity in those lines had a significant impact on their cancer-related phenotypes, we overexpressed in four of them (RNS85, 87, 96 and 130) either IRE1 WT or the variant Q780* (which impairs IRE1 signaling as seen in Fig 2 of the original manuscript in U87 cells). These data are now presented in the **new Figure 8** of the revised manuscript. These data are now further documenting the role of IRE1 signaling in GBM tumor phenotypes and provide information on the respective roles of the IRE1/XBP1s and RIDD axes.

We strongly believe that the data added to the manuscript make our story much stronger and for this reason we would like to thank again the reviewers for their great help for pointing towards the weaknesses of our initial work.

2nd Editorial Decision

05 December 2017

Thank you for your patience. We have finally received the enclosed two reports from the referees asked to re-assess it. As you will see the reviewers are now globally supportive and I am pleased to inform you that we will be able to accept your manuscript pending the following final amendments:

1) Please address in writing the comments of referee #4 and include a point-by-point response to this referee and to my comments.

***** Reviewer's comments *****

Referee #3 (Comments on Novelty/Model System for Author):

New changes are fine

Referee #3 (Remarks for Author):

The authors took all comments very seriously and addressed all my concerns. I am positively surprised by the number of changes and inclusion of numerous novel assays that improved the article dramatically.

Referee #4 (Comments on Novelty/Model System for Author):

Authors have utilized cell line and patient derived models of cancer for this study.

Referee #4 (Remarks for Author):

Overall, this revised version of the manuscript is a significant improvement over the last version. Authors have attempted to address almost all the comments/critiques from reviewer. However, there is still some ambiguity in how data is analyzed and presented to the reviewer, at least in mind of this reviewer. Some of the major issues in the manuscript are listed below:

1. It is still not clear to the reviewer why authors are so adamant on utilizing microarray data when RNAseq data from TCGA could significantly improve resolution of XBP1/RIDD signature. Specifically, IRE1 is involved in splicing and authors can look in to differences in splice variants when RNAseq data is used for analysis.
2. It is still not clear why authors emphasize so much on importance of mutation in IRE1 when it is clearly the case of IRE1 over activation in GBMs (as evidence by no mutation in GBM TCGA cohort - Most recent analysis in cbiportal).
3. Authors have provided some details on how XBP1s/RIDD signature classification was derived.

However, when heat maps in Figure 6B & C and Figure 7A are compared, it is not clear how could RNS175 classify as XBP1+/RIDD+? (XBP1s Score ~1.5 and RIDD score ~2.0).

2nd Revision - authors' response

08 December 2017

1) Please address in writing the comments of referee #4 and include a point-by-point response to this referee and to my comments.

1. It is still not clear to the reviewer why authors are so adamant on utilizing microarray data when RNAseq data from TCGA could significantly improve resolution of XBP1/RIDD signature. Specifically, IRE1 is involved in splicing and authors can look in to differences in splice variants when RNAseq data is used for analysis.

We thank this reviewer for his/her suggestion to investigate RNAseq data. Using these data, we were able to positively correlate the XBP1s signature and the presence of sequence reads corresponding to the spliced form of XBP1 mRNA. This has been carried out in the manuscript and can be seen on the revised figure 6 and S7. This is indeed true that IRE1 participates to mRNA splicing, however this happens through a non-conventional, cytosolic and spliceosome independent splicing mechanism. The precise mechanisms underlying IRE1 dependent mRNA splicing are yet to be fully characterized and thus it might be very difficult to figure out what type of event is fully dependent on IRE1. Furthermore, IRE1-mediated splicing has thus far been exclusively limited to XBP1, and thus the amount of work necessary i) to overcome the dogma and ii) to sort out IRE1-mediated splicing from all splicing events stands far beyond the scope of our manuscript.

2. It is still not clear why authors emphasize so much on importance of mutation in IRE1 when it is clearly the case of IRE1 over activation in GBMs (as evidence by no mutation in GBM TCGA cohort - Most recent analysis in cbiportal).

The revised version of the manuscript no longer emphasizes the potential role of the mutations in IRE1. We have used this information as a tool to further explore the signaling pathways downstream of IRE1 through RIDD or through XBP1 mRNA splicing.

3. Authors have provided some details on how XBP1s/RIDD signature classification was derived. However, when heat maps in Figure 6B & C and Figure 7A are compared, it is not clear how could RNS175 classify as XBP1+/RIDD+? (XBP1s Score ~1.5 and RIDD score ~2.0).

We thank this reviewer for this remark, it was indeed a mistake from our part in the previous version of the manuscript that has now been fixed in the revised version of the manuscript.

Corresponding Author Name: Eric Chevet

Journal Submitted to: EMBO MOLECULAR MEDICINE

Manuscript Number: EMM-2017-07929